# Pioneers of the Ice Age Models: A Brief History from Agassiz to Milankovitch

M. Efe Ateş[1]

[1]Department of Philosophy, MSKÜ, Mugla, 48000, Turkey

*Correspondence to*: M. Efe Ateş (mefeates@mu.edu.tr)

**Abstract.** It is currently known that astronomical factors trigger the emergence of glacial and interglacial periods. However, nearly two centuries ago, the overall situation was not as apparent as it was with today's scientists. In this article, I briefly discuss the astronomical model of ice ages put forward between the 19th and 20th centuries. This century was indeed *anni mirabiles* for scientists to understand the ice age phenomenon. Agassiz, Adhémar and Croll laid the foundation stones for

understanding the dynamics of ice ages. But it was Milankovitch who combined empirical geology with mathematical astronomy. To put specifically, he identified the shortcomings of the preceding ice age models and modified his model accordingly. In what follows, I review former approaches to the ice age problem and show how they failed to meet their objectives. Next, I show how Milankovitch's model managed to capture all sufficient astronomical elements. Last sections focus on Milutin Milankovitch's genuine approach, including his accomplishment of tackling the problem mathematically.

**1 The Problem**

There seems no strict quantitative definition of ice ages (see Davis, 2001; Barry and Yew Gan, 2011, p. 299). However, we may easily say that ice ages are periods when severe temperature reduction occurs across the Earth. During such periods of time, surface of the Earth is largely covered with ice sheets and mountain glaciers. So far, scientists provide evidence that the Earth, in its history, has experienced many glacial periods. However, only few of these have established dominance over the

whole climate system by lasting more than millions of years. The chart below shows these extensive ice ages in the past 2.4 billion years (Fig. 1).

As seen from the chart, scientists have identified five significant ice age periods during the past 2.4 billion years. They are respectively called Huronian (2.4–2.1 billion years ago), Cryogenian (850–630 million years ago), Andean–Saharan (460–430 million years ago), Karoo (350–260 million years ago) and Quaternary (2.6 million years ago–present). Accordingly, we

are living in the Quaternary period which began approximately 2.6 million years ago. Since we live in the Quaternary period, we are currently in the last major ice age period. So, the Earth should be in a notably colder climate compared to our current condition. At present, however, this is not the case. Neither climate is extremely cold, nor are our continents largely covered by ice sheets. On what grounds, then, scientists claim that we are in a major ice age period?

Geologists already has the answer to this question. It seems now the received view that the Earth currently is in an
interglacial period that approximately have started about 15 thousand years ago (Lisiecki and Raymo, 2005). According to
this view, our planet, sometime during an extensive ice age period, makes alterations toward warmer climatic conditions.
After witnessing warmer climate conditions, it again goes into a deep freeze and this specific event recurs at intervals. Thus,
in short, the Earth's climate is subject to certain quasi–periodic changes. With respect to time, these alterations are small
steps for the Earth, but giant steps for humankind.

At first sight, the intuitive way to evaluate the alterations between glacial and interglacial periods is to assume a probable
mechanism behind it. The scientists did so as well. Whether periodic or not, the Earth has witnessed, and probably will
continue to witness numerous glacial and interglacial periods. Thus, for scientists, the new task is to find out which factors
are responsible for this. The common questions which they had dealt with were as follows: Why the Earth's climatic
conditions change within major ice ages? What factors are involved in the separation of glacial and interglacial periods?
Why series of glacial and interglacial intervals are either longer or shorter than some other ones? The remainder of this
article will be concerned with these questions and the answers given to them, but before proceeding let me say few words
about the occurrence of major ice ages.

## 2 Major Ice Ages

There exist many causes of major glaciation; including oceanic fluctuations, volcanic eruptions and surface albedo (see also
Rohling *et al*., 2012). Without the feedback mechanisms initiated by plate tectonics, however, long term oscillations of
climate would be completely different. So to speak, the main cause responsible to initiate an ice age period is plate tectonics.
Alfred Wegener (1880–1930), the German geologist and meteorologist, has laid the groundwork for this idea. He spent his
time primarily in Greenland and his field research was –to a certain extent– centered upon continental drifts that later led
Harry Hess (1906–1969) to develop the revolutionary theory of plate tectonics. The theory of plate tectonics was not
specifically proposed with the purpose of explaining the mechanism of major glacial periods. Nonetheless, it provided a
useful framework to explain the dynamics of long term climate changes.

Given this theory, the separate continents today were once combined together. For instance, the well–known supercontinent
Pangaea was one of them. It existed approximately between 350–260 million years ago and different from today's
positioning, its continental mass was mostly located in southern hemisphere of the Earth. The east part of South America and
the west part of Africa were bind together where India was located to the southeast of them. After a considerable time, each
of these landmasses broke apart and drifted away until reaching their current positions (Fig. 2).

This theory has much to say about geological phenomena. With its principles, for example, we can deduce hypotheses about
other natural events, such as volcanic eruptions, earthquakes or processes of mountain formation. Eventually, all these
phenomena are results of the movements of plates which are segments of the continents and the oceans.

Similarly, the beginnings and the ends of major glacial periods are also related with the movements of continents. Many relevant textbooks give a clue about this relationship by using the term 'continental glacier' because glaciers (whether they are ice sheets or ice caps) can only form on ground. This information is crucial for the formation of glaciers, but it is also crucial for the explanation of the initial conditions of glacial periods. In order to understand this, it's sufficient to consider that the positions of continents have decisive influence over the global climate. To put it precisely, it happens as follows:

When the continents are positioned mostly near the poles, low amount of sunlight falls on a big portion of total continental area. This means that snow and ice accumulate over large areas. Hence, the albedo increases and significant part of sunlight returns back to space. As a result, global temperature decreases and ineluctably produces a glacial period. This is why, many researchers have asserted that "the centering of a land mass on the pole or grouping of land masses around a pole to be an essential condition for glaciation" (Hay 1996, p. 410).

This natural process can be exemplified by reviewing the structure of the supercontinent Pangaea. As I mentioned above, large portion of Pangaea continent was located close to the south pole of the Earth. This means that suitable conditions were provided for a glacial period to start. Today, based on paleoclimatological evidence, we know that the Earth experienced an extensive ice age about 350–260 million years ago, also the era when Pangaea existed.

At this point, it is especially important to note that the theory of plate tectonics has no room or explanation for the questions

raised at the end of the previous section. It is because this theory accounts for the long–term changes of global climate and falls short of establishing short–term temperature variations in major ice ages. According to the theory of plate tectonics, plates move so slowly (1–1.5 centimeters per year on average), and due to this fact, continents which are parts of plates move slowly too. This basic mechanism of the drift is extremely gradual such that continents shift their positions over millions of years. So, any model deduced from the theory of plate tectonics appears to be capable of explaining long term

climatic changes. However, it would unavoidably fail to provide an explanation of what factors may be responsible for our climate to warm up and regress back to a cold climate within major ice age periods. So, the factors that lead to major ice ages do not properly explain short–term climate changes. To find out the causes of the short–term climate variations within major ice ages, then, we must narrow down the time scale from millions of years to thousands of years.

## 3 An Astronomical Solution for a Geological Puzzle

As described above, climatic changes within the ice age periods are not caused by movements or positioning of continents. For this reason, we need to seek the causes of these temperature changes, elsewhere, in the Earth system. Milutin Milankovitch (1879–1958), the Serbian mathematician, astronomer and engineer, thought so and came up with an idea to connect climatic changes with Earth's orbital variations. According to him, temperature changes during the major ice age periods depend on the different amounts of solar irradiance falling on the surface of the Earth. By all means, the amount of

solar irradiance is driven not by the Earth's internal system but by an external force, namely the Sun. Therefore, a possible solution lies not only in the field of Geology, but also in the field of Astronomy.

The idea to understand the dynamics of ice ages from the astronomical perspective is not new, and hence it does not belong entirely to Milankovitch (see Andersen 1992; Hestmark 2018; Berger 2021, p. 1731). There are other details which make Milankovitch an original figure in science. These details and his other contributions will be spelled out in more detail after.

But first, I will outline briefly the pioneers of the astronomical/geological theory who influenced Milankovitch's approach to the ice ages one way or another.

## 4 Pioneers of the Ice Age Theory

We know quite a lot about the glaciers and glaciation, such as how ice sheets move, how glaciers shrink and grow, or how glacial deposits are formed etc. The situation, however, was different over the past two centuries. In those days, few

scientists were involved with the glaciation research. Moreover, many were not aware how crucial this natural phenomenon is in shaping the solid surface of the Earth.

The first sign of the ice ages was erratic boulders. These erratic boulders (large masses of rock) are found in places far from their bedrock source. Early to mid–18th century, the received view was that these giant rocks are somewhere distant from their original places because they are transported and deposited by a great flood –as almost told in the Bible. Although he

changed his mind later on, Charles Lyell (1797–1875), the British geologist, confidently defended such a view and accepted it as the most likely explanation for the emplacement and movement of boulders (Imbrie and Imbrie, 1979, pp. 21–22, Boylan 1998, pp. 156–158). While this view dominated the geological scene for over decades, it subsequently received criticisms from some researchers in the scientific community. The main objection urged against the flood theory was that such giant rocks are less likely to be carried from low to high elevations via floodwater. So, there must be another

mechanism at work to transport massive boulders from one place to another. In addition, there was one other thing that made it hard for flood theorists to account for: scratched and grooved bedrock surfaces. These traces on the land surface were an indication of a serious problem because flood theorists were assuming a simple process of transportation of boulders from one place to another. However, in order to erode the underlying land surface such a way, boulders must strongly scrape against the ground that they move over. Therefore, floodwater alone was not a possible candidate to be the cause of

scratched and grooved surfaces.

The time when the level of the erratic boulder discussion manifested itself in a puzzling way, Louis Agassiz (1807–1873), the Swiss biologist and geologist, was working on fossil fishes. However, along with fossil research, his mind was busy with another issue: the possible causal link between erratic boulders and glaciation. According to Agassiz, the boulders which we find far from their bedrocks are in their current different places not because they are carried by the flowing water but because

they are removed by an agent with much stronger force. His hypothesis was that the displacement of a boulder from its bedrock could most probably be produced by the effect of glaciers. Especially, for Agassiz, polished bedrocks were a tangible sign of an ice age that had taken place in the distant past. As he himself stated:

These surfaces are sometimes even, sometimes undulated, often traversed by furrows more or less deep and sinuous, but which never occur in the direction of the slope of the mountain. On the contrary, these furrows are oblique and longitudinal; in short, they have a direction which at once excludes the idea of a current water having been the cause of the erosions… To any one who has seen the Alps, it is evident that it is the ice which has produced this polishing (Agassiz 1838, p. 177).

To tell the truth, this hypothesis was not entirely original (see Seylaz 2017; Davies 2017). The relationships between boulder displacement, polished bedrocks and glaciation were suggested by many travelers and scientists prior to Agassiz (see Berger 2012, pp. 108–110). Given the history of science, however, this is not new news. It is a rare possibility that a scientific claim has not been expressed before in a similar way by any other scientist. However, there are different and even worse allegations about the Agassiz's case. He has been accused of intellectual dishonesty for failing to disclose that his hypothesis was borrowed from Karl Schimper (1803–1867), a German botanist (see Evans, 1887). I will not delve into this story further because as the Latin proverb says *errare humanum est*. Although he allegedly did not cite or credit relevant previous work, we should emphasize or recognize his admirable attitudes and attributes. The initiative he took in the organization of the Swiss Society of Natural Sciences, for example, is worth appreciating.

In 1837, Agassiz gave a talk about his glacial theory to the Swiss Society of Natural Sciences at Neuchatel. The expectations have failed, because members of the society were prepared to hear a talk about his new research results on fossil fishes. Instead, Agassiz took the opportunity and announced his new argument on the phenomenon on glaciation. Truly, from that moment onwards the dispute had begun. Within the time, Agassiz was exposed not only to the criticisms of opponent camps, but also showed resistance to the career–oriented advices from the authorities of geology, including William Buckland (1784–1856) and Alexander von Humboldt (1769–1859). For example, von Humboldt once wrote in a letter:

I am afraid you work too much, and (shall I tell you frankly?) that you spread your intellect over too many subjects at once. I think that you should concentrate your moral and also your pecuniary strength upon this beautiful work on fossil fishes. In so doing you will render a greater service to positive geology, than by these general considerations... In accepting considerable sums from England, you have, so to speak, contracted obligations to be met only by completing a work which will be at once a monument to your own glory and a landmark in the history of science... No more ice, not much of echinoderms, plenty of fish... (Agassiz, 1886, pp. 267–272).

Despite such tough conditions, only about 30 years after, the theory suggesting the link between erratic boulders and glaciation was widely accepted in the scientific community, and this constituted a milestone in the advancement of glaciation research.

After being established as a physical process that occurred once or many times in the past, glaciation became a subject of geological investigations. As Agassiz stated in a letter to his friend, the glacial periods are hence regarded "by geologists as a fixed fact" (Agassiz 1871/1949, p. 294). Now the leading issue was no longer about the existence of past glacial periods. Instead, it was about the initial conditions of glaciations.

The French astronomer and mathematician Joseph Adhémar (1797–1862), was the first prominent scientist who drew attention to the onsets of glaciation process. According to him, temperature changes (like glaciation–interglaciation) in both hemispheres occur due to the precession of the equinoxes. In his book *Révolutions de la Mer*, he argues as follows:

[B]ecause of the precession of the equinoxes, the sums of the number of hours of day and night in each
       hemisphere are unequal. This inequality produces a difference in the corresponding temperatures, and it is
       this difference that explains the origin of the ices at the two poles (Adhémar, 1842, p. 96).

As stated by Adhémar, the durations of day and night differ in each hemisphere (except two days of the year, the spring equinox on March 21 and the autumnal equinox on September 23). The axial tilt of the Earth is the key factor underlying this
difference. Today, the north pole of the Earth points to Polaris, more commonly known as the North Star. However, this will not always be the case. For Adhémar, the axis of the Earth wobbles and changes orientation between pointing at Polaris and Vega with a cyclic period of 22.000 years. In other words, the North Star will be Vega about 11.000 years later. Because the orientation of the axis changes very gradually, the Earth's tilt remains almost unchanged over the period of one annual year. Whichever hemisphere tilts towards the Sun gets more light and hence experiences warm seasons. For instance, the northern
hemisphere tilts toward the Sun as the Earth begins to move away from the Sun (March 21–September 23). Conversely, the southern hemisphere tilts toward the Sun as the Earth begins to move closer to the Sun (September 23–March 21).

According to Adhémar, one other point needs to be considered as well i.e. Earth's elliptical path around the Sun. The information that the Earth orbits the Sun in an ellipse, not a circle had already been known by means of Keplerian laws since the seventeenth century. The first one of these laws states that the Sun is not located exactly at the center of the ellipse, but
rather located at one of the two foci. When the Earth gets closer to the Sun, the speed of our planet increases due to gravitational motion. The orbital speed of the Earth affects the number of days in each season. For example, the cold seasons (winter–autumn) of the northern hemisphere are short in number of days. On the other hand, when the Earth moves away from the Sun, the gravitational pull becomes weaker and our planet travels on its orbit at a relatively slow speed. As the Earth's orbital motion slows down, the northern hemisphere experiences warm seasons (summer–spring). Thus, the warm
seasons of the northern hemisphere are more in numbers of days. Therefore, in the northern hemisphere, the number of days in the warm seasons is more than the number of days in the cold seasons.

According to Adhémar, the difference in the number of days between the two seasons exactly corresponds to 168 hours, i.e. 7 days. The situation is reversed, however, for the southern hemisphere. "The southern [hemisphere] will… lose, in a year, more heat than it receives, since the total duration of its nights exceeds that of the days by 168 hours" (Adhémar 1842, p.
27). If we take for unity, for example, the average amount of solar radiation that the Earth receives in 1 hour, then we find – at the end of the year– 336 units of irradiation difference between the two hemispheres. Following this calculation, Adhémar concluded that the northern hemisphere enjoys longer summers, while the southern hemisphere experiences longer winters. Because the southern hemisphere experiences longer winters, it is actually in the process of glaciation.

In truth, Adhémar's views have been highly criticized. Many researchers, especially his contemporaries, "did not want to believe that the southern hemisphere is on average much colder than the northern hemisphere" (Bard 2004, p. 632). Despite these criticisms however, Adhémar still deserves to be remembered as an elite figure in the history of science, for laying the foundations of the astronomical theory of ice ages.

Another remarkable attempt to understand the onsets of glacial periods was made by the Scottish scientist and geologist James Croll (1821–1890). Croll was aware of Adhémar's work, and was also sympathetic to some parts of it. He, for example, believed that long and cold winters would lead to glaciation in the corresponding hemisphere, as Adhémar suggested. Nevertheless, he was not inclined to ascribe particular importance to the precession of the equinoxes as a cause of glaciation. As we can recall, Adhémar was correct in stating that the northern hemisphere has long warm seasons, while the southern hemisphere has long cold seasons. In this fashion, he concluded that the northern hemisphere was experiencing an interglaciation, since it receives a great deal of insolation annually. On the other hand, the southern hemisphere was experiencing a glaciation due to opposite circumstances. However, Croll thought that this conclusion is wrong because if a particular hemisphere, for example, receives less sunlight in cold seasons, it would in turn receive more sunlight in warm seasons. Thus, the amount of heat loss in the winter season will be compensated by the following summer season in both hemispheres. As Croll puts,

> Whatever extra heat the southern hemisphere may at present receive from the sun daily during its summer months owing to greater proximity to the sun, is exactly compensated by a corresponding loss arising from the shortness of the season; and, on the other hand, whatever daily deficiency of heat we in the northern hemisphere may at present have during our summer half–year, in consequence of the earth's distance from the sun, is also exactly compensated by a corresponding length of season (Croll, 1875, p. 82).

According to Croll, the compensation of heat loss or heat gain would almost nullify the effect of the axial precession. In this regard, no significant climatic impact would occur in each hemisphere. Thus, he reasoned that glacial periods must occur in periods when the heat compensation fails between cold and warm seasons. Axial precession alone might be a necessary factor, but apparently it was not sufficient. Therefore, there must be another factor at play to prevent the heat compensation mechanism from functioning. Croll identified this factor as high eccentricity. This also was the very idea which separates Croll further from Adhémar. As stated above, Adhémar knew that the Earth revolves around the Sun in an elliptical orbit. Nevertheless, he neglected the fact that this orbital shape might slightly change in time. In other words, he took eccentricity fixed at some value between 0 and 1 (circle and parabola, respectively). According to Croll, on the other hand, the modulation of axial precession by eccentricity should be considered because the mutual heat compensation between seasons could fail only in such condition. In this sense, he stated that we should take the combined influence of these two causes into account,

> There are two causes affecting the position of the earth in relation to the sun, which must, to a very large extent, influence the earth's climate; viz., the precession of the equinoxes and the change in the eccentricity of the earth's orbit. If we duly examine the combined influence of these two causes, we shall find that the

northern and southern portions of the globe are subject to an excessively slow secular change of climate, consisting in a slow periodic change of alternate warmer and colder cycles (Croll 1864, p. 129).

By means of two orbital factors, Croll explained the conditions in which the seasonal temperature difference could not be compensated for a given hemisphere. The hemisphere which tilts away from the Sun at times when the Earth is located at the far end of the elliptical orbit with a high eccentricity, it would likely to be in a process of glaciation. Croll claims that these two factors are responsible for unstable conditions of the seasons in a given hemisphere; nonetheless, they are not the *direct* causes of glaciation. Put it more precisely, it is the Earth's physical agents that produce glaciation processes, not factors such

as orbital forcing. These physical agents comprise many phenomena from the Earth's surface to Stratosphere, including clouds, fogs, wind–driven ocean currents etc. Croll, for example, described the effects of certain meteorological factors on glaciation as follows:

> Under a cloudless sky, the direct rays of the summer–sun would … be more than sufficient to remove the winter's accumulation of ice and snow. But if from thick fogs or an overcast sky the direct rays of the sun
were prevented from penetrating to the earth, the heat of summer would not in such a case be sufficient to remove the snow and ice; and the formation of glaciers would be the inevitable result (Croll 1864, p. 133).

Among the phenomena that Croll identified as physical agents, the most noticeable one is albedo effect. It is also noteworthy that no scientist until his time drew attention to this phenomenon (see Bol'shakov, Kapitsa and Rees 2012; Thompson 2021). Albedo effect is a mechanism that triggers the glaciation by reflecting the solar radiation back to space. As glaciers spread

across landmasses, they cause the Earth to absorb lesser heat. In due course, the average temperature of our planet starts decreasing. This, in turn, creates a positive feedback loop which promotes the further spread of glaciers till they reach their maximum level. Here, one may easily identify the circular process. However, in addition to being circular, this process is also self–reinforcing. The effect amplifies the further effect of its initial cause, or to sum up in Croll's own words, "cause and effect mutually react so as to strengthen each other" (Croll, 1875, p. 75).

As it seems, Croll attributes the primary cause of glaciation to physical agents (e.g. albedo, ocean circulations). Put differently, the Earth's orbital motions only set the stage and conditions for physical agents. Nevertheless, it is worth mentioning that for Croll, there is no fundamental distinction between causes and conditions. This view was expressed in his highly philosophical work, published a few weeks before he died:

> The distinction between cause and conditions is to a great extent arbitrary. There is no real or essential
distinction between the two … [A]ll the conditions are co–operating causes; and the selected one, which we term the cause, is effective only in co–operation with the others (Croll 1890, p. 68).

I believe this quote is key to understanding Croll's general view on the relation between glaciations and orbital motions. In this sense, physical agents (like albedo) are directly responsible for causing great climatic changes; however, orbital motions of the Earth are responsible for determining the appropriate initial conditions. As he states: "an increase of eccentricity

could have no direct tendency to lower the temperature and cover [a] country with ice, [but] it might bring into operation physical agents which would produce this effect" (Croll, 1875, p. 13).

    Agassiz, Adhémar and Croll are three significant figures that influenced the course of the glaciation research. However, they are not the only ones who deserve all the respect. There are many other scientists who contributed substantially to the field of ice age studies, by mainly observing nature over the years and making hypothesis about the past, present and future of the

environmental conditions. Particularly, Imbrie and Imbrie (1979), Bard (2004), Berger (2012) and Paillard (2015), in their seminal and detailed works, traced the history of great glaciologists, their naturalist approaches and the breakthroughs in glaciation research. In the following section, however, I focus on Milankovitch whose contribution had a significant impact on the science of glaciation among all those researchers.

## 5 The Astronomical Model of Ice Ages

Like many scientists, Milankovitch stood on the shoulders of giants as well. The contribution he made to science, particularly to climatology was only possible with the past contributions of great scientists. In truth, Milankovitch often mentions the names of his predecessors in his book *Kanon der Erdbestrahlung und seine Anwendung auf das Eiszeitenproblem*, not only to proceed with their past research work, but also to give credit to their scientific legacy.

    Nevertheless, as much as the past efforts should be recognized, we should also emphasize Milankovitch's genuine approach

to the problems. His approach is different and genuine from most of his predecessors because it involves some sort of unifying framework. Until the era of Milankovitch, the mainstream methodology on particular issues in geology was dominantly descriptive. However, for Milankovitch exact methodology (or in other words, mathematical approach) should be integrated into this descriptive stance. For him, only if this condition is met, we could hope to discover the geological traits of the Earth and other planets.

The time when Milankovitch encountered the problem concerning the glacial and interglacial periods, he was already equipped with this line of thought. According to him, those who were dealing with the issue had the necessary information to proceed, but they were not able to achieve it. Of course, there have been reasons for that. Given Milankovitch's interpretation, some of the scientists had little idea what to do with empirical evidence and how to relate them with relevant theory; while the others were unable to use their theoretical knowledge to construct the model of the investigated

phenomenon (see Imbrie and Imbrie, 1979, pp. 97–99; Petrovic, 2012). In all this, Milankovitch was defending a unified view that suggests a combination of both approaches. Although, scientists like Croll made similar attempts to unite these approaches, they inevitably failed due to lack of empirical evidence or their insufficient mathematical training (see Milankovitch, 1941, p. 376).

    As stated above, the way Milankovitch approached the problem was quite original. He argued that crucial climatic changes,

such as series of glacial and interglacial intervals, are causally connected with the distribution of solar radiation that reaches the Earth's surface. Thus, any change in the amount of insolation triggers a change in the global climate. So, in order to

understand much about the periods of glacial and interglacial climates, we must determine *all* the factors that vary the amounts of insolation.

According to Milankovitch, the distribution of incoming solar radiation happens in accordance with the Earth's orbital variations. These orbital variations cause little or no change in the amount of total insolation that reaches the Earth's surface, but they do play an essential role in seasonal changes. For example, they change the seasonal duration (long or short winters/summers) or the degree of difference between the seasons (hotter, milder or cooler winters/summers). Moreover, these orbital variations comprise three astronomical cycles: orbital eccentricity, axial tilt and precession. Most commonly they are called 'Milankovitch Cycles' in deference to the Serbian scientist.

## 6 Milankovitch Cycles

I have briefly discussed two orbital elements (axial precession and orbital eccentricity) within the context of Adhémar's and Croll's approach. Nevertheless, there is a third orbital element called obliquity in addition to these two. In this section, I will briefly explain how each of these three elements operates in a cyclic fashion. The orbital movements of the Earth are referenced multiple times throughout the *Kanon*. Nevertheless, it is in the fourteenth chapter that Milankovitch represents all three cycles in the form of a compact diagram (Fig.3).

In the diagram "S represents the centre of the [Sun], and the ellipse PIAIIIP [represents] the annual terrestrial orbit round the Sun" (Milankovitch 1941, p.246). The line SV drawn through S is normal to the orbital plane and "the angle VSN represents the inclination of the rotational axis of the Earth or the obliquity ε of the ecliptic" (Milankovitch 1941, p. 246). P and A respectively denotes the perihelion and the aphelion. Finally "the angle IIISP, which may be called ∏γ, represents the longitude of the perihelion to the vernal point" (Milankovitch 1941, p. 246). Figure 3, in its original form, perfectly illustrates the dynamics of the Earth's orbital motions but to provide further understanding, let me deconstruct the diagram into its three components and focus on each cycle separate from the others. (Fig. 4)

The first component of orbital variations is called orbital eccentricity. It can simply be described as the Earth's orbital path around the Sun. The gravitational tug of other large planets influences Earth's orbit. This leads to a change in orbital shape and hence produces a cycle lasting about 90000–100000 years. Actually, our planet appears to orbit around the Sun elliptically. However, the shape of this orbit is not fixed. Sometimes it becomes more elliptical and sometimes it becomes more circular. The orbit becomes more elliptical when the difference between the furthest and closest distance of the Earth from the Sun increases. Differently, when the difference between the furthest and closest distance of the Earth from the Sun reduces, the orbit becomes more circular. The place where the Earth is nearest to the Sun is called perihelion, which occurs around January 3. On or around July 4, however, the Earth is at its greatest distance from the Sun, which is called aphelion (Fig. 4a).

Orbital eccentricity takes a value obtained by a simple formula using the variables aphelion (a) and perihelion (p) as follows: $e = (a - p)/(a + p)$. If the obtained value for orbital eccentricity is equal to zero (e=0), the shape of the orbit is a perfect

circle. If it is greater than zero (e > 0), then the shape of the orbit is an ellipse. When the current estimated values are inserted

into the above equation, we get $e = 0.016$.

Orbital eccentricity has an influence on climate change. However, it is the least effective factor on glaciation, among other variations. The reason is that the eccentricity variations have small impact on total annual insolation, namely a difference of 0.03% (see Maslin and Ridgwell, 2005, p. 21). Nevertheless, orbital eccentricity of the Earth may lead to significant temperature contrast between seasons. For example, if the orbital path of the Earth around the Sun was circular, there would

be no annual insolation difference between summer and winter for each hemisphere. But, due to gravitational effects of other celestial objects, especially Jupiter's, this cannot happen. As stated, today, our planet's path is elliptical in shape, approximately with a value of 0.016. Therefore, the difference of seasonal insolation between summer and winter reaches to about 6%. So, this shows that the amount of solar radiation reaching to the Earth's surface is greater at perihelion, for northern hemisphere. If the Earth's orbital eccentricity is at its maximum value (0.07), namely the most elliptical shape,

difference of seasonal insolation between summer and winter would reach to about 30%. As a result, seasons would be at their extremes in terms of temperature (e.g. very hot summers) for one hemisphere and would be moderate (e.g. milder summers) for the other.

The second component of orbital variations is called axial tilt or obliquity. Our planet's rotational axis is tilted relative to its orbital plane. The angle of the tilt is determined by drawing a line perpendicular to the Earth's orbital plane. The angle of the

tilt changes between 21.5° to 24.5° on a 41000 year cycle (Fig. 4b)

Today, the obliquity of Earth is measured as 23.5°. This tilt can be measured easily at solstices and equinoxes. In order to do that, it is sufficient to take the inverse tangent of the value which is found by dividing an object's height by its shadow length, at that particular time. When the angle of tilt is about to increase its maximum value 24.5°, the temperature contrast grow sharper between the two seasons. In such a case, winters become colder and summers become warmer. Contrarily,

when the angle of tilt decreases, the characteristics of seasons come closer. In this case, milder winters follow cooler summers.

The third and last component of orbital variations is called precession of the equinoxes. While orbiting the Sun, our planet also wobbles on its axis, like a spinning top. This wobbling of the Earth on its axis periodically repeats itself every 26.000 years. Today, the polar axis of the Earth points to Polaris, also known as the North Star. In fact, this won't last forever. Due

to wobbling motion, the Earth's axis will gradually change and it will point to Vega. More specifically, around 13.000 years from now, Vega will be the North Star and 13.000 years after that Polaris will once again be the North Star (Fig. 4c). Thus, in effect, it takes 26 000 years for the axis to return to the same spot on the circle.

The main effect produced by precession is an alternation of the seasons for each hemisphere. For example, today, the northern hemisphere is in winter, the southern hemisphere is in summer at perihelion. An opposite situation will be observed

by the middle of the cyclic period. To put it differently, when the Earth's rotational axis points to Vega, the northern hemisphere would be in summer and the southern hemisphere would be in winter at perihelion. In such a situation, the summers become warmer and the winters become colder. In other words, seasonal contrast in temperature appears stark.

While the information of orbital cyclicity was not new at all for many astronomers, there was no available calculation of past orbital variations until Urbain Le Verrier (1811–1877), the French astronomer and mathematician. By applying Newtonian

laws to the masses of planets, he calculated the past changes of orbital motions of the Earth. In this way, he created a sort of data table that displays the past variations of Earth's orbit in the last 100000 years. After decades, U.S. American astronomer John Nelson Stockwell (1832–1920) calculated the changes in obliquity and eccentricity of eight planets in the solar system. Furthermore, in 1904, German mathematician Ludwig Pilgrim (1844–1927) extended the calculations further and provided data on orbital changes of the Earth for the last million years.

Having these data is a big step forward for Milankovitch because they will be used in revealing the relationship, if there is any, between insolation amount and global climate change of the Earth. However, Milankovitch was extremely careful and meticulous in his endeavors. He knew that the changes in solar irradiation are causally connected with the three orbital variations. He was also perfectly aware that these variations could be determined by precise calculation of all the planets' masses and of their motions. So, had they not inserted the actual masses of the planets while calculating the past orbital

variations, the amount of insolation received by the Earth could not have been calculated correctly for remote past (see Milankovitch, 1941, p. 252–253).

Milankovitch firstly made use of the calculations done by Pilgrim. The reasons were two–fold. First, Pilgrim's calculations covered the past 1 million years of the Earth's orbital variations. Using a calculation with such a wide time interval would certainly make things easier in understanding the past insolation amounts of the Earth. Second, Pilgrim's calculations were

done including the times when Earth's longitude of perihelion was equal to 90° and 270°. The importance of this is that, according to Milankovitch, great climate changes tend to take place at times when the terrestrial perihelion attains the values of 90° and 270°.

Pilgrim was a good mathematician, or so Milankovitch had heard (see Milankovitch, 1941, p. 253). But regardless of how he was talented in mathematics, he made use of Stockwell's integrals. The work of Stockwell was not entirely negligible, but it

did contain many printer's errors, as well as some calculation mistakes which firstly pointed out by German mathematician and astronomer Paul Harzer (1857–1932). In such a condition, all that remained was the calculations of Le Verrier's. Compared with the works of Pilgrim and Stockwell, Le Verrier's calculations were better, but unfortunately they were also lacked in accuracy. Milankovitch, in a sense, was stuck for an answer and did not know what kind of strategy is best for his purposes. Eventually, critical support came from his colleague Vojislav Miskovitch (1892–1976), the Serbian astronomer.

Using Le Verrier's integrals, Miskovitch did all the necessary re–computation by correcting the mass values of the planets (see, Janc *et al.*). This was thought to be a highly effective way because it would allow them to make a reasonable comparison, particularly between Le Verrier's recomputed results and Stockwell–Pilgrim's results. If the results indicate a slight difference, then the Earth's past insolation values could reliably be derived from these –otherwise things could get really tricky. Luckily, this strategy took hold in a significant manner. "The comparison of the insolation values that

Milankovitch calculated from these solutions show[ed] a good agreement" (Berger 2021, p. 1728). After settling these

preliminary points, the remaining task was essentially to combine the orbital variations of our planet with a mathematical model.

Milankovitch was well aware of the past astronomical theories and their shortcomings in explaining the phenomenon of glaciation. For example, Adhémar correctly emphasized 26.000 years (however, Adhémar actually thought it was 22.000 years) cycle of precession as one of the most responsible factor for glaciation but considered the orbital eccentricity as a constant. Hence, this incorrect thought, no doubt, led him to conclude that ice ages could occur only in one hemisphere. In other words, his theory suggested that one hemisphere would experience ice age condition while the other would be ice–free during a period of 13.000 years (again, for Adhémar it was 11.000 years). Croll did not make the same mistake and paid much attention to the Earth's orbital eccentricity. Furthermore, he was the first who noticed the idea of albedo, or the reflectiveness of the Earth's surface. Although the effect of albedo is decisive to some extent, unfortunately that idea led him to think that colder winters would influence the expansion of glaciers. Thus, apart from neglecting the effect of obliquity, he insisted that ice ages are driven by very cold winters at aphelion.

According to Milankovitch, both earlier theories were incomplete because they do not contain all necessary parameters. In other words, either one or two orbital parameters are used in these early theories to give an explanation for glaciation. However, all parameters have an effect on the glaciation, one way or another and they should all together be taken into account. In order to be a significant global change in the climate, all the cyclic variations of the orbital motions must be in a state of superposition (see Milankovitch, 1941, p. 271). Because all three cycles operates independently "sometimes their influence on the amount of heat received by certain parts of the Earth nullifies each other" and "sometimes the changes increase or decrease the quantity of the heat" (Grubic, 2006, p. 199). The complex nature of orbital variations makes it difficult to express them mathematically, and it was precisely this difficulty that plagued Milankovitch's predecessors. According to him, both early approaches were mathematically incapable of linking the effects of orbital variations on insolation. He states this explicitly in his book *Kanon* as follows:

> All these theories have … the same shortcoming: None of them have correctly grasped the variability of all the astronomical elements which affect the irradiation of the Earth … [and] besides, none of these theories was able to tackle mathematically the decisive influence of variations in the obliquity on the irradiation of the Earth (Milankovitch, 1941, p. 376).

And then continues as follows:

> I was able to show that the astronomical problem of ice ages is far more complicated than had been assumed before, and that in order to arrive at a correct solution the whole problem had to be approached fundamentally and put on a broad basis (Milankovitch, 1941, p. 376).

Milankovitch's fundamental approach was to base his work on a mathematical model. Before constructing the model, he set some preliminary ground for a concept, namely the canonic unit (see Grubic, 2006, p. 199). Early on his research, Milankovitch noticed something interesting and important about climatological works. Until his time, no one had attempted to propose a mathematical theory of climate. One of the main reasons for this was that the majority of climatologists gave

prime importance to empirical studies. The confidence in their instruments (with which they could make accurate measurements) was far greater than their confidence in a theory (which may turn out to be inaccurate). Another reason was related to a natural phenomenon, namely the Sun's rays. Nearly everybody knew that the Sun is the Earth's largest source of radiant energy, and that also its rays pass through the atmosphere. What was unknown was whether the strength of these rays could be expressed as a constant.

U. S. American astrophysicist Samuel P. Langley (1834–1906) was the first scientist who initiated the research on solar constant at the Astrophysical Observatory of Smithsonian Institution. Langley's aim was to measure the mean solar radiation amount with the bolometer, an instrument he had invented. In this way, the Sun's rays could be expressed in calorie per square centimeter per minute. In 1884, Langley calculated the solar constant radiation corresponding to 3.1 grams–calories per cm² per minute (see also Langley, 1903). Almost thirty years after, however, in the yearbook of the same institute,

scientists announced that they determined the new value of solar constant corresponding to 1.946 gram–calories per cm². Although Langley was not able to establish solar constant precisely, this unit is named langley (Ly) to honor his legacy. Milankovitch was aware of these studies at the Smithsonian Institution (see Milankovitch, 1941, p. 306). Thus, he knew that the mean value of solar constant is found to be 1,946 grams–calories per cm². It was clear, for Milankovitch, that decades were spent to obtain the solar constant value, nevertheless, it could not be determined precisely. So, "this value [1,946

grams–calories per cm²] is not yet to be considered as final" either (Milankovitch, 1941, p. 213). It seemed reasonable for Milankovitch not to use a value whose precision has not yet been established. For this reason, he expressed the mean value of intensity of the solar radiation in canonic units, and considered canonic unit ($J_o$) as the solar constant. He took canonic unit "as the unit of radiation, and a hundred thousandth part of the year as the time unit" (Milankovitch, 1941, p. 266). To put it in symbols, $J_o=1$; T= 100.000.

It may seem perhaps arbitrary to express average solar radiation in canonical units, but actually it is not. According to Milankovitch, using the canonical unit would be advantageous to compute the course of insolation, i.e. winter and summer half–year insolation ($W_W$, $W_S$) and thus total year insolation ($W_T$). In his words, "in computing the secular variations of $W_W$, $W_S$, $W_T$ and the secular march of terrestrial irradiation, it will be particularly advantageous to express insolation in question in canonic units" (Milankovitch, 1941, p. 266). To illustrate this computational advantage, it is sufficient to refer to two

separate tables in *Kanon*, particularly Table VII and Table XI. In both tables, the insolation values of $W_W$, $W_S$, and $W_T$ are expressed in canonical units for the latitudes from 0° to 90°. In these tables, it is not specified to which hemisphere a particular latitude belongs to because, according to Milankovitch, for example, 65° N and 65°S receive the same amount of irradiation in their northern and southern summer half–years, respectively.

$W_W$, $W_S$, and $W_T$ of the particular latitude differ between the two tables because they are produced with the same equations but different coefficients. Table VII, for instance, show the total year radiation quantities of 65° by using the heat unit $J_o = 2$ (notice that, this value is very close to 1.946) and the time unit 365, 2422 days, i.e. 525.942 minutes. Thus, the coefficient $TJ_o$ in Milankovitch's initial insolation model is 1.051898 (2x525.942). On the other hand, Table XI indicates $W_T$ of 65° in terms of canonic units, so $TJ_o$ in that model is 100.000 (1x100.000). When these calculations are made, by taking into account the relevant orbital parameters, the insolation amount for 65° in Table VII and Table XI is respectively as follows:

Table VII 65° ($W_S$=143.000, $W_W$=22.180, $W_T$=165.180)

Table XI 65° ($W_S$=13.594, $W_W$=2.109, $W_T$=15.703)

All the values shown in Table VII can also be obtained from Table XI. To do this, it is necessary to divide the coefficient value in the equation used for table VII by the coefficient value in the equation used for Table XI (=1.051898/100.000). When the obtained number 10.519 is multiplied by $W_T$ of 65° in Table XI, we get approximately $W_T$ of 65° in Table VII, i.e. 165.180 (10.519x15.703=165.179).

All these calculations actually show one thing: Milankovitch's simplifications are valid. The time unit is not the actual number of minutes in a year (525,942 minutes). Rather, a year is divided into 100,000 equal parts. The value of heat unit is not 1.946 grams–calories, but is accepted as $J_o$=1. Despite all these simplifications, however, the insolation model using the canonical unit is in great agreement with the model that contains actual units. Moreover, the simplified model reduces the calculation labor as well.

After having settled all these matters to his satisfaction, Milankovitch presented his mathematical model for insolation. The essential elements for the model are as follows: summer and winter half–years for a certain year, changes of irradiation at a certain latitude for summer and winter, the change of the inclination of the ecliptic, eccentricity of the ecliptic in the given year, longitude of the perihelion relative to equinox and the coefficient for individual latitudes.

So, the model comprises all the necessary components including three orbital parameters, the changes of irradiation amount for specific latitudes and the coefficient that expresses the insolation amount in caloric units. With all these, the mathematical model would allow us to calculate the secular march of insolation for both the northern and southern hemispheres.

To understand how much solar radiation that the Earth received in the past thousands of years, we need to know what the current insolation value is. If the model can accurately provide the current insolation values, by relying on the actual values of the orbital parameters, then, its results can be extended to represent the past and as well as the future climatic conditions of the Earth. Thus, the retrodictive and predictive inferences about the Earth's climatic patterns would be valid if and only if they are deduced from true statements, i.e. from verifiable or factual statements. In light of these considerations, Milankovitch believed that his mathematical model needs to be verified or tested through empirical evidence. For this reason, he decided to calculate the mean annual temperature of the lower layer atmosphere for various latitudes. If the calculated results significantly agree with the empirically measured data, then the model would pass a test of verification.

Weather stations, as they do today, were regularly measuring the mean annual temperatures at various latitudes. Observations of weather conditions were recorded and transformed into data outputs. Milankovitch planned to compare the results of his model with this available data at his time. Focusing particularly on the northern hemisphere, he performed certain calculations. By means of the model, he assessed trends in average temperatures for northern latitudes. The average annual temperature values of the northern latitudes between 40°–50° were almost the same as the average temperatures recorded by the weather stations. However, there were significant temperature deviations at latitudes from 40° N to the equator and from 50°N to the north. Milankovitch, for example, found the average temperature of 10° N to be higher than the value determined by the weather station. The opposite held also for the upper latitudes of the northern hemisphere. This situation in fact weakens Milankovitch's model, rather than verifying it.

Nevertheless, Milankovitch anticipated the factors that led to the difference or failure: air and ocean currents. He reasoned that air and ocean currents carry canonic units from one place to another. Therefore, canonical units do not actually disappear; so they are (or they should be) stored somewhere in the Earth's atmosphere. Accordingly, Milankovitch revised his approach to the topic, by calculating the mean temperature of the lower layer atmosphere for all the Earth's latitudes. The new results are compared with those of weather stations. The mean value difference between Milankovitch's model versus meteorological measurements was only 0.1° C. To put it clearer, this very slight difference indicated that Milankovitch's model is also verified or at least highly supported by empirical evidence.

With the Milankovitch model of insolation, it became possible to calculate how much solar radiation is received for any latitude in the past, present and future. Nevertheless, the model as such was not sufficient to understand the ice ages. To understand, completely, how the mechanism of ice age works, Milankovitch needed to know which seasons are crucial in triggering the glaciation process. The early theories were centered on very cold winters and accepted such severe winters as a decisive factor in accelerating glaciers (see Imbrie and Imbrie, 1979, p. 104). Milankovitch was skeptical about this hypothesis. However, being untrained in geological dynamics, he was far from abandoning this idea and putting an alternative approach. Yet, he was aware that he "had to solve" this "preliminary question of principal importance: which of the meteorological elements and which season was to be selected in the Ice–Period" (Milankovitch, 1941, p. 414).

In fact, the solution to this preliminary problem was proposed already by many scientists of the time. At the beginning of the 19th century, the German geologist and paleontologist Leopold von Buch (1774–1853) laid the foundation for this answer in Scandinavia, where he conducted his research on climatology. A main conclusion drawn from this research was the causal relationship between snow line levels and summer temperatures. This conclusion was in conflict with the common view that winters determine the snow line level. Nearly five decades later, Scottish physicist and glaciologist James David Forbes (1809–1868) made similar observations in Norway and supported Buch's claims with the following statement: "[observations support] the excellent generalization of von Buch, that it is the temperature of the summer months which determines the plane of perpetual snow" (Forbes 1853, p. 206). Shortly afterwards, Irish scientist Joseph John Murphy (1827—1894) maintained Forbes' line of thought by asserting "the idea that a long, cool summer and a short, mild winter are the most favorable conditions for glaciations" (Berger, 2021, p. 1729). As Murphy himself put, "we have plenty of observed data; and

I think I can show that they all go to prove a cool summer to be what most promotes glaciation, while a cold winter has, usually, no effect on it whatever" (Murphy, 1869, p. 351). Nevertheless, it seems very likely that Milankovitch was unaware of these views and was in need of an urgent intellectual help.

Eventually, the expected help came from the Russian–German geographer, and climatologist Wladimir Peter Köppen (1846–1940). He was aware of the works of Milankovitch and had read the monograph which is about the model of insolation. Soon after, he contacted Milankovitch. At the time, Köppen was researching on climate classification and climates of geological past with his son–in–law Wegener. Both researchers were experienced in the field, and they were familiar with almost all the relevant geological record as well as the seminal studies on ice ages. On a weekly basis, Köppen and Milankovitch exchanged dozens of letters. Finally, the discussions ended up, when Köppen convinced him that the decisive factor on glaciation was milder summers, not colder winters as usually thought. That did seem reasonable to Milankovitch.

> After an exhaustive discussion off all the possibilities, Köppen answered the question by indicating that it is the diminution of heat during the summer half–year which is the decisive factor in glaciation… I therefore used Köppen's advice and directed my attention to the periods of cold summers (Milankovitch, 1941, p. 414).

Truly, the suggestion to focus on milder summers was genuine because the key factor in the development of glaciation was about whether or not glaciers are preserved. At the high latitudes of the northern hemisphere, large amount of snow were accumulating and the glaciers were being formed even in completely different past climatic conditions. However, when the summers were hotter in any hemisphere, the glaciers formed in the winter were starting to melt slightly. Therefore, while milder or relatively colder summers were preserving glaciers and hence culminating glaciation; hotter summers were setting the stage for interglacial periods (see also Oerlemans, 1991).

Milankovitch had already constructed a model that makes it possible to calculate the insolation amount for a particular given latitude of the Earth. In this early version of the model, the strategy was to determine the difference between present and past values of insolation. The same strategy will also be followed in the revised model. However, the initial model was constructed on the basis of astronomical half–years. An astronomical year (or tropical year) is measured generally from either vernal equinox to the next one; or from the autumnal equinox to the next one. Following this measurement, the lengths of the seasons are considered as consisting of two parts. For instance, the northern hemisphere experiences astronomical summer half–year from 21 March (vernal equinox) to 23 September (autumnal equinox), and astronomical winter half–year from 23 September to 21 March. According to Milankovitch, however, the astronomical calendar was inappropriate to follow the secular march of insolation precisely because the length of each half–year in a particular year varies due to the variations of astronomical elements, especially due to the orbital eccentricity. Thus, this parameter should be modified correctly. In this manner, he divided "the year into two equally long seasons, one of which –the caloric summer– comprises

all days during which the irradiation at the given latitude is stronger than on any day of the other half–year, i.e. the caloric winter" (Berger 2021, p. 1729).

As it is understood, astronomical half–years are not proper means to demonstrate variations in past climate. Caloric–half years were introduced merely for this reason. Nevertheless, dividing astronomical year into two caloric half–years is in a sense problematic. To deal with this problem, Milankovitch offered two solutions, one analytical and one geometrical. In the following, I will discuss the latter solution (see Fig.5) –those readers who are interested in analytical solution should refer to *Kanon* (1941, pp. 275-278).

In the graph, we have the representation of the mean radiation w of a surface element at any arbitrary latitude (φ) as a function of time. Insolation is plotted along the vertical axis (w), while time is plotted on the horizontal axis (t). The points on the curve PFGQHKSF' represents the course of radiation at the latitude (φ). The points I,II, III, IV and I' on the timeline, respectively denotes the time of the vernal equinox, the summer solstice, the autumnal equinox, the winter solstice and the subsequent equinox. The astronomical summer half–year (Ts) is represented by the segment I–III, and the astronomical winter half–year (Tw) is represented by the segment III–I' (see Milankovitch 1941, 271–294).

To understand this solution, it would be sufficient to notice the plotted points QH and SF' on the curve. These are the last irradiation intervals of astronomical summer half-year ($T_S$) and astronomical winter half-year ($T_W$), respectively. When we compare QH with SF', however, we see that the final irradiation interval of $T_S$ (i.e. QH) is smaller than the final irradiation interval of $T_W$ (i.e. SF'). Like in the astronomical calendar, each half–year's duration is 182 days, 14 hours and 54 minutes, but differently a caloric winter–half year (½T on the right–hand side) comprises all the days receiving relatively less irradiation than any of the day in summer–half year (½T on the left–hand side).

Using the latest version of his model, the version in which both caloric half–years and canonic units included, Milankovitch (with the suggestion of Köppen) calculated past insolation amounts for the specific latitudes, particularly 55[th], 60[th] and 65[th] parallels of the northern hemisphere. He obtained the intended results over months and transformed these into fictitious latitudinal oscillations. The reason for that was to see a geometrical picture of the past summer insolation variations for the northern latitude of 65°. The created graph was an equivalent graph which showed the equivalent values of summer insolation amount for 65° N throughout the past 600.000 years (Fig. 6).

As seen in the graph, the past equivalent insolation values of 65° N are shown by curves for comparison. For example, about 115.000 years ago, the summer insolation amount of 65° N was almost equal to the today's summer insolation amount of 74° N. This could be evaluated as an indication of a relatively cold summer for the northern hemisphere. In the opposite cases of lower equivalent geographical latitude, the same latitude would of course have relatively warmer temperature averages in the summers.

Soon after completing his work, Milankovitch sent this radiation curve graph to Köppen. The graph must have been intriguing for Köppen, as it was in agreement with the findings of two German geographers Albrecht Penck (1858–1945) and Eduard Brückner (1862–1927). Penck and Brückner were researching on Alpine glaciers and about 15 years before Milankovitch's work, they identified four great glacial periods in Earth's history by examining successive gravels, plant

remains and moraines in the European Alps (Anderson, Goudies and Parker, 2013 p. 8). They displayed these different glacial periods in a graph and named them chronologically as Günz, Mindel, Riss, and Würm in their 1909 book *Die Alpen im Eiszeitalter* (Fig. 7).

The close agreement of the two graphs was a sign of victory for Milankovitch. Moreover, with this new graph, Milankovitch
provided a more precise graph compared to the old one. As seen in figure 6, the nine striped low points of the curve display colder summers which correspond to glacial periods occurred in the 589[th], 548[th], 475[th], 434[th], 231[st], 187[th], 116[th], 72[nd], and 22[nd] Millennia BP. Günz, Mindel and Riss, each includes two, and the last significant cold period Würm includes three low points. Clearly, Penck and Bruckner scheme does not have this preciseness.

**7 Conclusion**

On Milankovitch's part, the goal had been finally reached. His work "based upon exact science" passed into "the sphere of the descriptive natural sciences" and made the "link between celestial mechanics and geology" (Milankovitch, 1941, xvi). In short, exact science and descriptive science met by means of geological research on the one side and astronomical computation on the other. Furthermore, by constructing a mathematical model based on the orbital variations of Earth, it became possible to trace past climatic variations and also to predict the future glacial or interglacial periods. For example,
the model predicts that, without human effects, interglacial period which we are still in may end about 50.000 years after (see Berger and Loutre, 2002; Gajic 2019, p. 235, and also Paillard 2010).

Although, in 1924, Köppen and Wegener confidently published radiation curves in their book *Die Klimate der geologischen Vorzeit*, some remained sceptical about the validity of Milankovitch's model until 1970's. In the following years, however, Hays *et al.* (1976) published an article on the close link between climatic changes and the Earth's orbital variations. Based
on the oxygen isotope records provided by deep ocean sediment cores, they came to the conclusion that orbital changes induced climatic change in the past 500.000 years. In their terms, orbital cycles of the Earth should be understood as pacemakers of the ice ages. Eventually, this could be counted as a clinching victory for Milankovitch's model. In other words, Milankovitch's work was not merely providing a basis for further studies on mathematical climatology; it was also reliable in estimating the relationship between orbital variations and climatic changes, in a robust manner.


**Data availability.** No data sets were used in this article.

**Competing interests.** The author declares that he has no conflict of interest.

**Acknowledgments.**

I would like to acknowledge the Anonymous Referee for her/his useful remarks and comments. I also remain grateful to Andreas Schmittner. His insightful and helpful comments did much to improve this article. Special thanks to Z. Bora Ön for

his careful reading of the article and for making a number of constructive suggestions that improved substantially the initial manuscript. I wish to thank also to Jan Mangerud for kindly reminding me about recent works on astronomical theory of ice

ages.

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

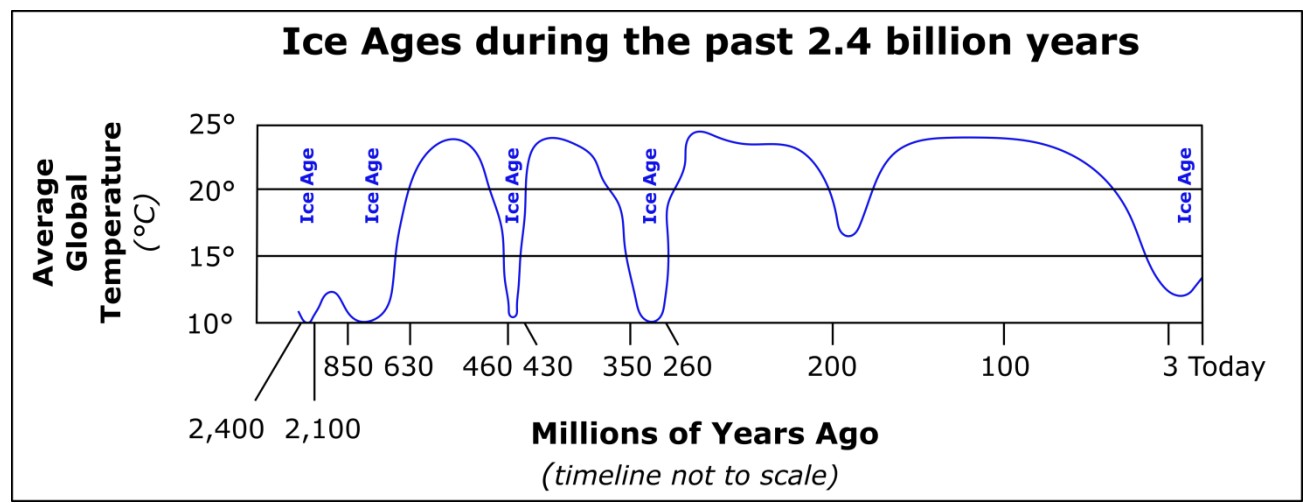

Figure 1: The chart shows five major ice ages during the past 2.4 billion years. Adapted from Utah Geological Survey Notes, Eldredge S. and Biek B. (2010) [Adapted from Saltzman (2002)], Retrieved from https://geology.utah.gov/wp-content/uploads/ ice_ages1 . © Copyright –Utah Geological Survey– State of Utah. May, 15, 2017.



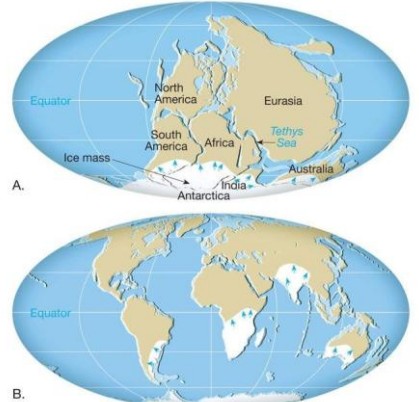

Figure 2: *A*. The supercontinent Pangaea existed approximately between 350–260 million years ago. Ice sheets are located close to the south pole of the Earth. *B*. Present position of the continents. Old ice sheet evidences are located on different land masses. Reprinted from Essentials of Geology (p. 282) by Lutgens, F.K., Tarbuck, E. J. and Tasa, D., 2012, U.S.A.: Prentice Hall. Copyright ©2012, 2009, 2006 by Pearson Education, Inc.



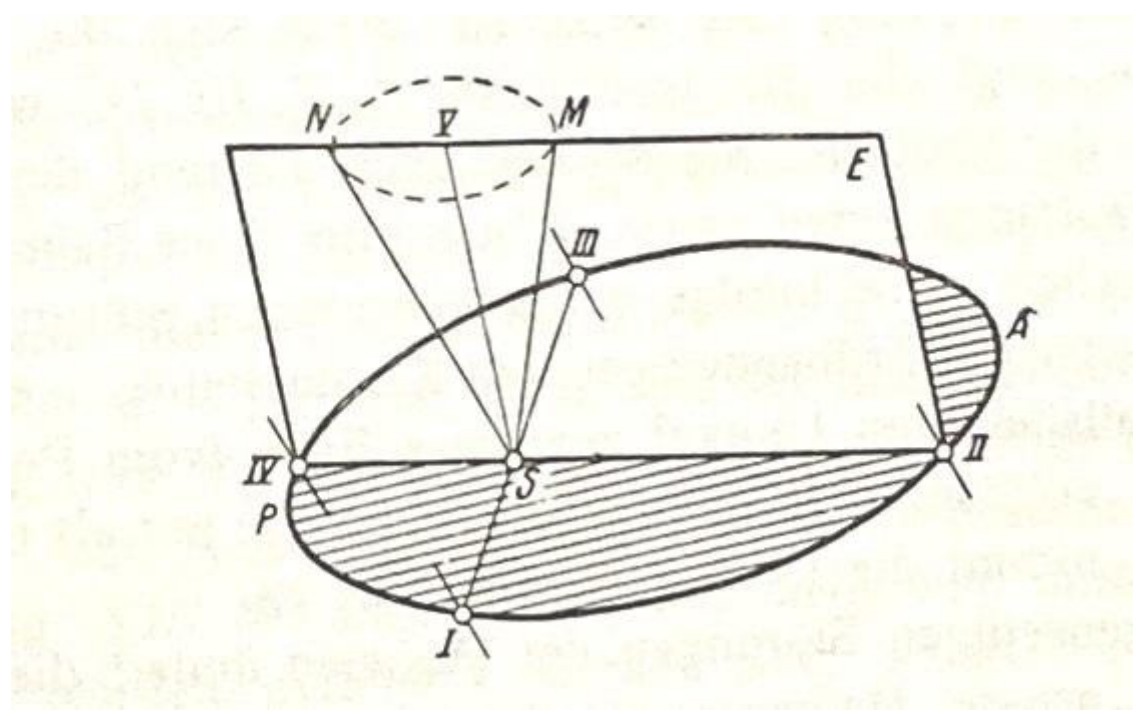

**Figure 3 The figure shows all astronomical elements that have an influence in modulating the amount of insolation.**
**Reprinted from *Kanon der Erdbestrahlung und seine Anwendung auf das Eiszeitenproblem* (p. 247) by Milankovitch, M., 1941, Belgrade: Royal Serbian Academy Special Publications.**


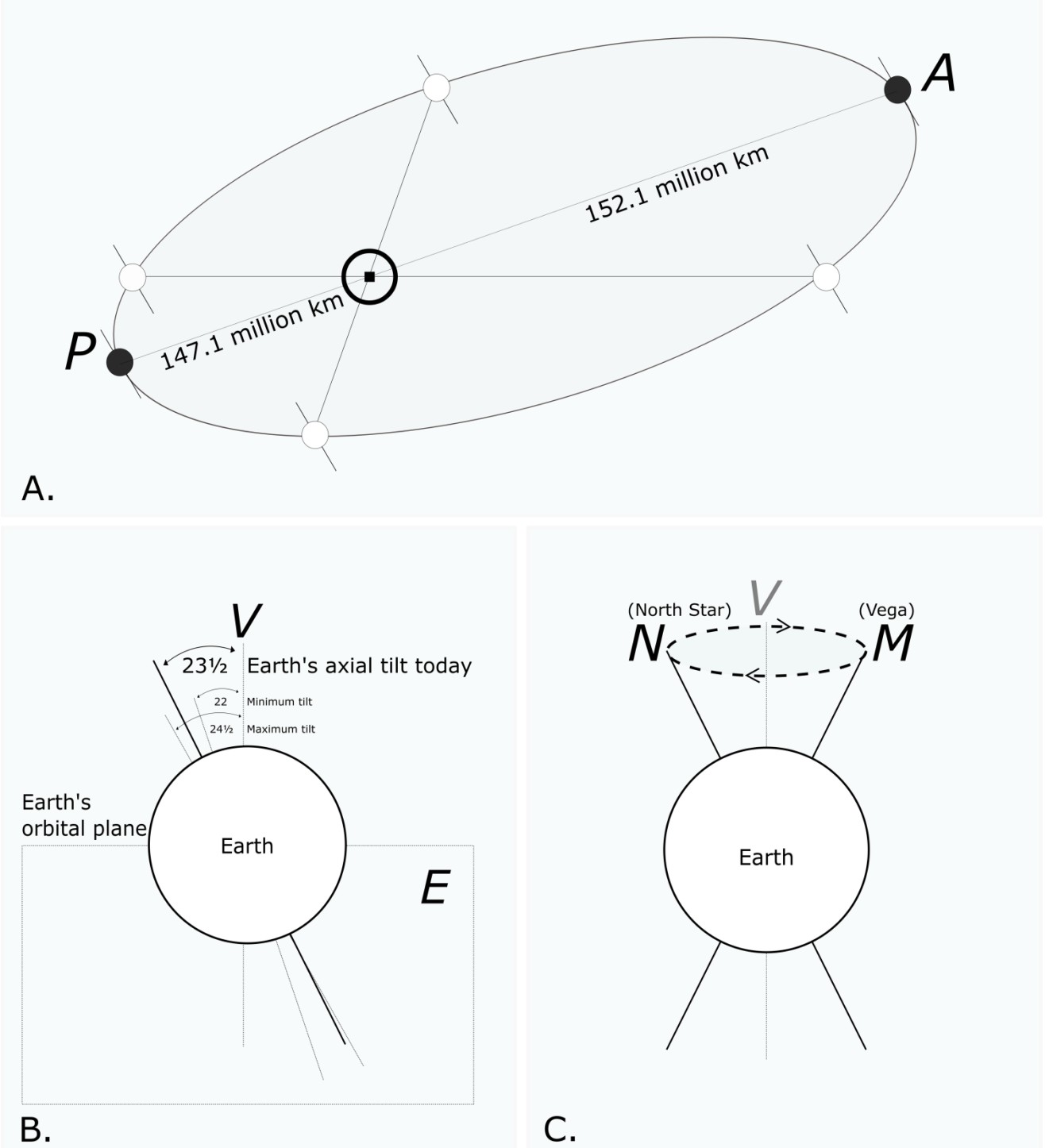

**Figure 4:** *A.* The distances from the Sun at perihelion and aphelion for the Earth today. *B.* The maximum, minimum and present tilt angles for the Earth. *C.* The wobbling motion of the Earth.

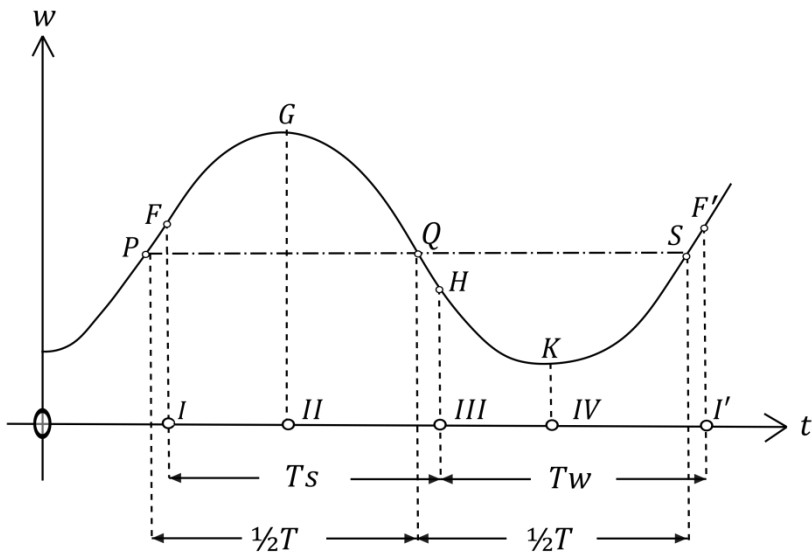

**Figure 5: The annual march of the irradiation for any arbitrary latitude. Reprinted from Kanon der Erdbestrahlung und seine Anwendung auf das Eiszeitenproblem (p.272) by Milankovitch, M., 1941, Belgrade: Royal Serbian Academy Special Publications.**

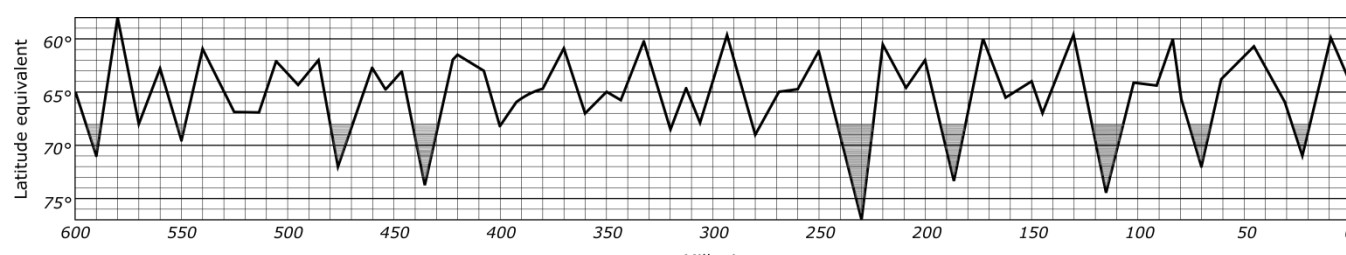

**Figure 6: The equivalent graph which shows the variations of the summer insolation for 65° N. Reprinted from Kanon der Erdbestrahlung und seine Anwendung auf das Eiszeitenproblem (p.415) by Milankovitch, M., 1941, Belgrade: Royal Serbian Academy Special Publications.**

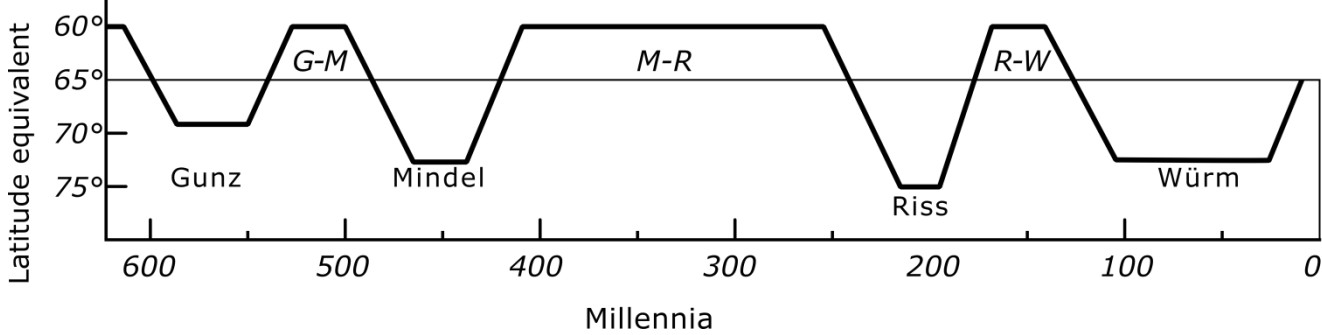

750

**Figure 7: Penck and Brückner's scheme of glacial/interglacial periods in the Alpines. Reprinted from Kanon der Erdbestrahlung und seine Anwendung auf das Eiszeitenproblem (p.417) by Milankovitch, M., 1941, Belgrade: Royal Serbian Academy Special Publications.**