# Peer review of "Pioneers of the Ice Age Models: A Brief History from Agassiz to Milankovitch"

_History of Geo- and Space Sciences, 2021_

## Community Comment (CC1)

**Comments on "Pioneers of the Ice Age Models: A Brief History from Agassiz to Milankovitch", by Ateş**

Z. Bora ÖN [*][1]

[1]*Muğla SK Üniversitesi, Jeoloji Mühendisliği Bölümü, Turkey*

January 21, 2022

**0    Foreword**

I have been invited to review this manuscript by the author, himself. I and Ateş are colleagues and friends, working at the same campus.

**1    General comments**

The manuscript presents a concise and adequate summary of the astronomical theory of ice ages. It first gives us the wider picture of previous ice ages and explains the plate tectonics and climate relation with examples. While the plate tectonics theory is newer, it constitutes an envelope for the long term climate change. Therefore, Ateş decides to start from the past ice ages of the world, and convinces the reader about why there were times that the world was ice-free. Then, the manuscript presents us the history of the discovery of Quaternary ice ages by starting from Agassiz. Then presents astronomical solutions proposed, namely, by Adhémar, Croll and Milankovitch, as a cause or for the oscillations of Quaternary ice ages. Then the article concentrates on Milankovitch's theory, its history and its outputs.

The article and presents us a different perspective for the subject. I believe, It is written well and it is adequate. My general comments are listed below;

[*][boraon@mu.edu.tr](mailto:boraon@mu.edu.tr)

1. The article gives the impression as the ice age concept is proposed by Agassiz. But, unlike Agassiz, the text should also give credit to Venetz, de Charpentier (see, Berger, 2012) or even to Hutton (see, Davies, 1968) and Playfair (see, Seylaz, 1962). Furthermore, mentioning Jens Esmark (Andersen, 1992, pp. 102) as the first scientist to propose an astronomical solution (even it is very primitive) would be better.

2. In the 6$^{\text{th}}$ section the astronomical parameters (lines between 201–254) are given with too much detail. They can be simplified. Furthermore, figures 3, 4 and 5 can be presented in a single figure.

3. The formulas between 2 to 9 distorts the integrity of the text. I believe these formulas and the paragraphs describing them should be rewritten from scratch just by referencing the original sources.

4. In text it is sometimes written *the earth* and sometimes *Earth*. It should be consistent. If I were Ateş, I would follow the recommendation of Şengör (2017) and use it as *the earth* not *Earth*.

**2 Line based recommendations**

Below you can find my specific recommendations.

- Line 38 (L38): Change "our climate is subject to certain periodical changes" to "the earth's climate is subject to certain quasi-periodic changes"

- L41-42: Change "Whether periodically or not, the Earth has witnessed, and probably will continue to witness numerous glacial and interglacial periods." to "Whether periodic or not, the earth has witnessed, and probably will continue to witness numerous glacials and interglacials."

- L45: "than their previous ones?" or "than some other ones?"

- L49: I would delete the first sentnce of this paragraph, starting with "There are many causes of major glaciation,...".

- L50: Consequently, I would change the sentence " Among all these, however, the main cause responsible to initiate an ice age period is plate tectonics." to "Without the feedback mechanisms initiated by plate tectonics, long term oscillations of climate (as illustrated in Figure 1) would be completely different."

- L51: A comma after meteorologist.

- L51: "this hypothesis", which hypothesis?

- L52: Change *had* to *was.*

- L53: This sentence is odd. The first part, till the comma, the reader understands it as Wegener's theory revealed the mechanisms of ice ages. Maybe it can be, "He spent his time primarily in Greenland and his field research was mainly focused on continental drifts that led him to develop the revolutionary theory of plate tectonics, which brought a useful explanation for long term climate changes."

- L58: Don't use "beneath", maybe "south" is better.

- L81: Delete "has".

- L82: Change "an extensive ice age about 350–250 million years ago, also the era when Pangaea existed. This can also be seen in (Fig .1), where one of the lowest points of the curve denotes this period." to "an extensive ice age about 350–250 million years ago, also the era when Pangaea existed (Fig .1)"

- L93: Referencing a first year textbook, Lutgens et al. (2012)... I am not sure if it is a good idea. Furthermore, this is the second time up to now and these are direct quotations.

- L124–L129: The story about Agassiz is not that innocent (Berger, 2012, pp. 109, 2$^{nd}$ column)

- L149: How would it be to change the wording here from "discovery" to "hypothesis"?

- L167: Comma after Milankovitch.

- L176: "Until the era of Milankovitch, the mainstream methodology on particular issues in geology was descriptive.", is this sentence necessary? Or, even true?

- L189–L193: "Milankovitch approached the problem was quite original.", but in this paragraph what you explain is similar to Croll's approach. What is original?

- L217: There is no need to show the computational line. The second line can be given in text, of course without that much precision, such as $e = 0.016$.

- Fig3: Please note that it is today's condition.

- L236: Are following sentences really necessary? "This tilt can be measured easily at solstices and equinoxes. In order to do that, it is sufficient to take the inverse tangent of the value which is found by dividing an object's height by its shadow length, at that particular time."

- L243: Yes, it is true that the precession of the equinoxes cycle lasts approximately 26,000 years. However, the climate is not affected by the sole effect of precession, but by the effect of the precession of the equinoxes modulated by the changes in the eccentricity of the earth's orbit. On average, its period is 21,700 yrs. Please see Berger (Table 1 in 1977).

- L245: Will Polaris once again be the North Star after 13,000 years or after 26,000 years? I suspect, there is a small mistake here.

- L255-261: It would be appropriate if you give original references of these studies.

- L275: I am not sure, but would it be OK to add that Croll was aware of the continental distribution of the earth?

- L321: "It is because all three cycles operates independently of each other.", I didn't understand the causality of being independent and being in a superposition of these cycles for a significant global climate change.

- L323: "When the quantity of the heat decreases, a glacial period begins. In the opposite situation, when the quantity of heat increases, the global temperature significantly rises up and an interglacial period begins consequently.", I don't think this is true. Insolation is not the marker of beginning of glacials and interglacials (cf. PAGES, 2016). I believe, these are all of Milankovitch's ideas or part of Milankovitch's model. They are not true for today. Therefore it should be explicitly emphasized.

- L395: "This could be evaluated as an indication of a relatively cold summer *for the northern hemisphere.*"

- L399: Is the following sentence correct? "Both scientists were already presented their results in a graph."

- L418: Is it true with human effects? Please check this. See also Fig. 6 of Paillard (2010) for possible future scenarios.

**References**

Andersen BG (1992). "Jens Esmark – a pioneer in glacial geology." *Boreas*, **21**(1), 97–102. doi:https://doi.org/10.1111/j.1502-3885.1992.tb00016.x.

Berger A (2012). "A Brief History of the Astronomical Theories of Paleoclimates." In A Berger, F Mesinger, D Sijacki (eds.), "Climate Change," pp. 107–129. Springer Vienna, Vienna. doi:https://doi.org/10.1007/978-3-7091-0973-1_8.

Berger AL (1977). "Support for the astronomical theory of climatic change." *Nature*, **269**, 44–45. doi:https://doi.org/10.1038/269044a0.

Davies GL (1968). "Early Discoverers XXVI: Another Forgotten Pioneer of the Glacial theory James Hutton (1726–97)." *Journal of Glaciology*, **7**(49), 115–116. doi:https://doi.org/10.3189/10.3189/S0022143000020451.

PAGES (2016). "Interglacials of the last 800000 years." *Reviews of Geophysics*, **54**(1), 162–219. doi:https://doi.org/10.1002/2015RG000482.

Paillard D (2010). "Climate and the orbital parameters of the Earth." *Comptes Rendus Geoscience*, **342**(4), 273–285. doi:https://doi.org/10.1016/j.crte.2009.12.006.

Seylaz L (1962). "Early Discoverers XV: A Forgotten Pioneer of the Glacial Theory John Playfair (1748–1819)." *Journal of Glaciology*, **4**(31), 124–126. doi:https://doi.org/10.3189/S002214300001827X.

Şengör AMC (2017). "Is it "the earth" or Earth?" *GSA Today*, **27**(3–4), 19. URL https://www.geosociety.org/gsatoday/archive/27/3/flip/i1052-5173-27-3-4/mobile/index.html#p=19.

---

## Author Response (AR2)

**Response to Reviewers and Commentator**

***hgss-2021-17*** *Submitted on 20 Oct 2021* ***Review article***
***Pioneers of the Ice Age Models: A Brief History from Agassiz to Milankovitch***
*Mustafa Efe Ates*

**Reply to RC1 (Anonymous Reviewer #1)**

Below are my responses/comments to the Reviewer 1. My responses/comments are in regular font, Reviewer's are in *italics*.

*General comments: This is a well written and enlightening article, which explains some pretty hard concepts in an easily understandable way, especially in the first half of the text regarding almost a century of scientific history. It is, consequently, a substantial contribution to science history within the scope of History of Geo- and Space Sciences. It also cleverly answers even the small questions it itself raises, such as: why do we not live in a notably colder climate compared to our current condition, since we live in the Quaternary, in an ice age period? (Answer: it seems that the Earth is currently in an interglacial period that started about 15 000 years ago, a blink for the Earth before it goes into a deep freeze again.)*

I would like to thank the Reviewer for reading my manuscript and also for the kind words. The Reviewer's general comments summarize this paper's aim in a concise manner.

*Specific comments addressing individual scientific questions/issues: The scientific and historical approach is valid, but the author may want to expand and deconstruct equations (6) and (7) to more elaborately and clearly explain the model before this early concluding, yet incomplete stage.*

I agree with the Reviewer's comment. Additionally, this comment is largely in agreement with the second Reviewer's concerns (Andreas Schmittner) and with the community commentator's (Z. Bora Ön) suggestions as well. Having all these in mind, I decided to eliminate the equations (from 2 to 9) entirely. To enhance the accessibility of the text for readers who want to be broadly informed about the issue, I will explain the equations in words. Consequently, part of the text (lines 300-380) will be rewritten accordingly in the final revised manuscript.

*Technical corrections – typing errors, etc: The results and conclusions are presented in a clear and concise way. The overall presentation is well structured. I could also not find any typing errors in the article, but the annus mirabilis should probably be anni mirabiles (for plural, in the sense of "marvelous years" or "years of wonders", since we are talking about a century or so).*

Done. I thank the Reviewer for the correction and the comment.

Below are my responses/comments to Andreas Schmittner (Reviewer #2). My responses/comments are in regular font, Reviewer's are in *italics*.

*This brief review of ice-age models is well written and I enjoyed reading it. I think it may provide a nice historical introduction for students and or non-climate-scientists to the topic. I especially liked the last two figures (7 & 8), which illustrate the applicability of Milankovitch's mathematical calculations with evidence from the geological record. However, I have two main comments/concerns that I think the author should consider in revising the work, in addition to several smaller and more technical comments.*

I would like to thank Andreas Schmittner for his kind words. I'm also very grateful for the valuable suggestions and for the careful reading of the manuscript. I think all his comments are to the point. I hope my replies here do justice to his comments and some of his concerns as well.

*My first main point is that a recent review of Milankovitch published last year by Andre Berger (https://doi.org/10.5194/cp-17-1727-2021) should be considered by the author. There is considerable overlap between the two works, and I would encourage the author to minimize the overlap, emphasize the differences, and to acknowledge the work by Berger.*

I thank the Reviewer for pointing out the work of Berger 2021and bringing it to my attention. Andre Berger is one of the main researchers in astronomical theory of paleo-climates. That is why I have cited his two different works in my manuscript. It is unfortunate that I was not aware of his 2021 article. This manuscript was written before 2021 and from that onwards I check new publications that may be of interest. Nevertheless, somehow I overlooked this relevant article. In sum, the Reviewer is certainly right when he says that "Andre Berger… should be considered by the author". Additionally, as the Reviewer constructively suggests, I will also minimize the overlaps and maximize the differences between the two works.

*My second main point is that the equations in the manuscript are not well motivated, and I didn't understand them. They should be better explained. E.g. in equations (2) – (5), what is the difference between Delta_Q and Delta_W? The different terms in the equations should be explained and motivated so that a typical reader can understand them.*

The Reviewer is right. I have received similar suggestions and comments from the Reviewer 1 and the Commentator 1 (Z. Bora Ön). Accordingly, the equations in the text will not appear in its original form.  In order to improve the tractability of the text for readers who want to be broadly informed about the issue, I will explain the equations in words. Accordingly, the part of the text (lines 300-380) which contain equations will be rewritten. In other words, you will find equations explained descriptively in the final revised manuscript. The Reviewer is also right when he says "I think it (the manuscript) may provide a nice historical introduction for students and or non-climate-scientists to the topic". My revised final version of the manuscript would be compatible with this saying as well.

*Detailed technical comments:*

*Line 35: replace "have" with "has"*

Thanks for the correction. Done.

*Line 107: replace "a glacier" with "glaciers"*

Thanks for the correction. Done.

*Line 220: insert "to" after "lead"*

Thanks for the correction. Done.

*Line 253: insert "at perihelion" after "winter"*

Thanks, it's worth emphasizing this detail. Done.

*Line 294: why is it compared here to the langley unit? Is this used elsewhere or was it a popular unit at that time?*

Langley unit [Ly] is a unit recognized by the International System of Units SI which is used to measure the heat budget. However, there are other units besides Ly. For example BTU (British Thermal Unit) and J/m² (joule per square meter) are some of them which are also used to measure the heat density (like incoming solar radiation, or forest fire). In effect, there is

no special thing about Ly. Nevertheless, by merely mentioning Ly leaves an opposite impression. So, following the Reviewer's comment, I will also convert Milankovitch's unit of radiation to other common units. This, I hope, may sweep away this opposite impression. I thank the reviewer for raising this point, which was not sufficiently specified in the manuscript.

*Line 298: replace "year's" with "half-year's"*

Done.

*Line 299: I assume "colder" here means "has less insolation than". Perhaps clarify.*

True. I will clarify this point in the revised version.

*Line 336-338: This is an interesting context, not mentioned by Berger's review.*

I warmly thank the Reviewer for this comment.

*Line 380: What for values other than 90 and 270? It would make more sense if the different signs represent a range of longitudes of perihelion.*

I thank Reviewer for the comment. Milankovitch gave paramount importance to $\prod\gamma$ (longitude of the perihelion relative to vernal equinox) because glaciation processes occurs when $\prod\gamma$ attains the values of 90° and 270°. In his words "The great amplitudes of insolation producing the glacial periods occurred… at times at which the longitude $\prod\gamma$ of the terrestrial perihelion attained 90° and 270°" (Milankovitch 1941, p.263). The situation with ↓ε (low obliquity), e↑ (large eccentricity) and $\prod\gamma$=90° favors the Northern hemisphere glaciation. The situation with ↓ε (low obliquity), e↑ (large eccentricity) and $\prod\gamma$=270°, on the other hand, favors the Southern hemisphere glaciation. That is why, in some other place, he emphasized Pilgrim's calculations as follows: "Pilgrim … computed the secular variations of the elements $\prod\gamma$, e, ε for each fifth millennium and for all those points of time at which $\prod\gamma$ equals 90° or 270°" (Milankovitch 1941, p.253). In responding to the Reviewer's question regarding values other than 90° and 270° (i.e. 0°, 180° and 360°), the best way perhaps is to cite Milankovitch again. He asserts the following: "When $\prod\gamma$ attains the value of 180° the annual seasons … are equal and both hemispheres stand on par […] At $\prod\gamma$ =360° both hemispheres, at equal annual seasons, are completely on par and everything starts again [0°]" (Milankovitch 1941, p.253). So, on December 21, for example, when $\prod\gamma$=270°, the below sign "-" is valid. It is because the mean insolation that the Northern Hemisphere receives (in southern winter) is less than the mean insolation the Southern Hemisphere receives (in northern winter). At $\prod\gamma$ = 90° the reverse conditions hold for two hemispheres. However, when $\prod\gamma$ =0°, $\prod\gamma$ = 180° and $\prod\gamma$ = 360° there would be no significant difference between both hemispheres with respect to the summer and winter (i.e. summer and winter lengths will be almost equal). Therefore, according to Milankovitch, the signs "+" and "−" are valid only at $\prod\gamma$=90° and $\prod\gamma$=270°.

*Line 395: insert "of lower equivalent geographical latitude" after "cases"*

Done.

*Line 399: Do we know if Milankovitch decided to plot this particular metric (equivalent latitude) for better comparison with work by others such as Penck and Bruckner?*

Thanks for this interesting question. I have no textual evidence that Milankovitch used a specific metric to compare his results with those of Penck and Brückner. Nonetheless, it seems intuitively plausible that Milankovitch made such an adaptation based on Köppen's advice. It is actually a quite nice topic to research upon. If I obtain any new information on this topic, I will include it in the final version of the manuscript.

*Fig. 1: units should probably be changed from deg F to deg C*

The units are now changed from °F to °C. Moreover, I have redrawn the chart because the first version was of poor image quality.

[Figure]

*Fig. 6: I don't understand this figure. Why is T_S not equal to 1/2 T? In other words, why does point F not coincide with point P? What is the difference? Please explain.*

The reviewer is certainly right. I sincerely thank the reviewer for making this point because this is a point which I did not sufficiently explain in the manuscript. TS and TW denote astronomical half-years, the summer and the winter half-years, in order. But for Milankovitch, TS and TW do not separate the year according to stronger and weaker insolation. Instead, TS and TW divide the year according to the duration of the day. However, as Milankovitch asserts, the duration of the day has no any significant relation with the insolation. The analytical solution is given by Milankovitch in the **Kanon** (1941, pp. 275-278). On the other hand, the figure that I used in the manuscript (fig.6 below) represents the geometrical solution.

[Figure]

In the figure, time is plotted on the horizontal axis (t). The insolation, on the other hand, is plotted along the vertical axis (w). The points on the curve PFGQHKSF' represent the course of radiation at an arbitrary latitude ($\varphi$). The astronomical summer half–year (Ts) is represented by the segment I–III, and the astronomical winter half–year (Tw) is represented by the segment III–I'.

To understand this solution, please kindly notice the plotted points QH and SF' on the curve. These are the last irradiation intervals of astronomical summer half-year (TS) and astronomical winter half-year (TW), respectively. When we compare QH with SF', however, we see that the final irradiation interval of TS (i.e. QH) is smaller than the final irradiation interval of TW (SF'). To avoid this unwanted situation, Milankovitch introduces caloric half-years. Like the astronomical calendar, each year's duration is 182 days, 14 hours and 54 minutes, but differently a caloric winter includes every day which is colder than the days in the summer half year (1/2T on the right-hand side). Conversely, a caloric summer includes every day which is warmer than the days in the winter half year (1/2T on the left-hand side). In this way, Milankovitch provides "a better insight into the march of insolation in remote times" (1941, pp. 274).

**Reference(s)**

Milankovitch, M.: *Kanon der Erdbestrahlung und seine Anwendung auf das Eiszeitenproblem*, Belgrade: Royal Serbian Academy Special Publications, vol. 132, (Canon of insolation and the ice-age problem, English translation by Israel Program for Scientific Translations, Jerusalem, 1969), 1941.

**Comments to the author**:
*I have a number of minor corrections and suggested slight changes in wording.*

I would like to thank Kevin Hamilton, handling topical editor, for his careful corrections and reasonable suggestions. I find that all his recommendations are on point. Below are my comments concerning the proposed changes; text that are in **bold** and *italics* character represent Kevin Hamilton's corrections and suggestions. My comments are in regular font.

*Line 6 "It is currently known that…" Replace with "It is now widely accepted that…"*

Thanks for the suggestion. Done.

*Line 7 "..as it was with today's scientists." Replace with "as it is today."*

Thanks for the correction. Done.

*Line 8 "between the 19th and 20th centuries." Replace with "in the 19th and early 20th centuries."*

Done

*Line 8 " "The century.." Replace with "This period …"*

Thanks for the suggestion. Done.

*Line 11 "To put specifically, …" Replace with "Specifically, …"*

Thanks for the suggestion. Done.

*Line 12 "..former approaches.." Replace with "…earlier approaches"*

Thanks for the suggestion. Done.

*Line 13 "Last sections…" Replace with "The final sections…"*

Done.

*Line 14 "..genuine approach…" Replace with "..successful approach.."*

Thanks for the suggestion. Done.

*Line 17 "…may easily say…" Replace with "will just say simply…"*

Thanks for the suggestion. Done.

*Line 18 "..surface of .." Replace with "The surface of…"*

Done.

*Line 20 "The chart below shows…..(Fig. 1)" Replace with "Fig. 1 shows…"*

Thanks for the suggestion. Done.

*Line 27 "Neither climate is extremely.." Replace with "The climate in not extremely…"*

Done.

*Line 28 "..then, scientists claim.." Replace with "…then, do scientists claim…"*

Thanks for the correction. Done.

*Line 29 "Geologist already has…" Replace with "Geologists already have…"*

Done.

*Line 29 "It seems now the received view that…" Replace with "It seems that the received view now is that…"*

Thanks for the correction. Done.

*Line 30 "…that approximately have started about 15 thousand .." Replace with "…that started approximately 15 thousand…"*

Done.

*Lines 33-34 "With respect to time, these alterations are small steps for the Earth, but giant steps for humankind." Replace with "On a geological scale these alterations are a normal progression, but for humankind a glacial-interglacial transition is a giant step."*

Thanks for the suggestion. Done.

*Line 40 "Why series of glacial and interglacial intervals either longer or shorter than some other ones?" Replace with "Why do series of glacial and interglacial intervals vary in length?"*

Many thanks for the suggestion. Done.

*Line 46 Delete "So to speak,"*

Thanks for the suggestion. Deleted.

*Line 52 "Given this theory.." Replace with "According to this theory.."*

Done.

*Lines 52-53 "For instance, the well-known supercontinent Pangaea was one of them. It existed approximately… Replace with "The well known supercontinent Pangaea existed approximately.."*

Done.

*Line 55 "..were bind together.." Replace with "were bound together"*

The correction is done. Thanks.

*Line 55 Delete "..of them."*

Thanks for the suggestion. Deleted.

*Line 65 "…low amount.." Replace with "…a low amount…"*

Many thanks for the correction. It is done.

*Line 66 "..significant amount…" Replace with "..a significant amount.."*

Many thanks for the correction. It is done.

*Line 77 "..move so slowly…" Replace with "..move very slowly…"*

Thanks for the suggestion. Done.

*Line 103 ""Early to…" Replace with "In the early to…."*

Done.

*Line 109 "..less likely to…" Replace with "..unlikely to…"*

Thanks for the suggestion. Done.

*Lines 100-111 "..was one other thing that made it hard for flood theorists to account for:…" Replace with "..was one other observation that flood theorists had difficulty accounting for:…"*

Done.

*Line 116 "The time…" Replace with "At the time…"*

Many thanks for the correction. It is done.

*Line 137 "…at Neuchatel. The expectations have failed, because members of the society were prepared to hear a talk…" Replace with "…at Neuchatel, although the members of the society were expecting to hear a talk…"*

Done.

*Line 140 Delete "Within the time"*

Thanks for the suggestion. Deleted.

*Line 140 "…opponent camps…" Replace with "…opposing camps…"*

Thanks for the suggestion. Done.

*Line 150 "Despite such tough conditions.." Replace with "Despite such skepticism…"*

Many thanks for the suggestion. Done.

*Line 150 "…30 years after…" Replace with "…30 years later…"*

Many thanks for the correction. It is done.

*Line 168 "..one annual year…" Replace with "..a single year…"*

Thanks for the suggestion. Done.

*Lines 185-186 Delete sentence "If we take for unit…between the two hemispheres."*

I follow Editor's suggestion. Deleted.

*Line 189 "In truth Adhemar's views have been highly criticized." Replace with "At the time Adhemar's view were widely criticized"*

Thanks for the suggestion. Done

*Line 191 "..an elite figure.." Replace with "…a leading figure.."*

Thanks for the suggestion. Done

*Line 237 "..is albedo effect." Replace with "…is the albedo effect.."*

The correction is done. Thanks.

*Line 239 "Albedo effect…" Replace with "The albedo effect…"*

The correction is done. Thanks.

*Line 240 "..lesser heat…" Replace with "…less heat…"*

The correction is done. Thanks.

*Line 259 "…..all the respect." Replace with "…credit."*

Thanks for the suggestion. Done

*Line 265 Delete "..as well."*

Deleted.

*Line 266 "In truth,.." Replace with "Indeed…"*

Thanks for the suggestion. Done

*Line 268 "…to proceed with their.." Replace with "..to reference their…"*

Done.

*Line 269 "genuine approach…" Replace with "..unique approach.."*

Thanks for the suggestion. Done

*Line 270 Delete "…and genuine…"*

Deleted.

*Line 274 "..met, we could.." Replace with "..met, could we …"*

Many thanks. Done.

*Line 352 "…appears stark." Replace with "..becomes intensified."*

Done.

*Line 353 "While the information of…" Replace with "While the notion of…"*

Thanks for the suggestion. Done

*Line 354 "…until Urbain…" Replace with "until the work of Urbain…"*

Thanks for the correction. Done.

*Line 360 "…is a big step.." Replace with "..was a big step.."*

Thanks for the correction. Done.

*Line 360 "..they will be used…" Replace with "…they were used…"*

Thanks for the correction. Done.

*Line 360-361 Delete ".., if there is any,"*

Deleted.

*Lines 373-374 "…p. 253). But regardless of how he was talented in mathematics, he made use of Stockwell's integrals. The work of Stockwell was not entirely negligible, but it did contain mainy printer's errors, as well as some calculation mistakes which was firstly pointed out by German mathematician and astronomer Paul Harzar.…" Replace with "…p. 253), but he made use of Stockwell's integral formulae which contained many printer's errors as well as calculation mistakes, as first pointed out by German mathematician and astronomer Paul Harzar.."*

The author would like to thank the Editor for his suggestion. Done.

*Lines 377-378 "..they we also lacked in accuracy." Replace with "..they were also somewhat inaccurate."*

Thanks for the suggestion. Done

*Line 389 "emphasized 26,000 years.." Replace with "emphasized the 26,000 years…"*

Thanks for the correction. Done.

*Line 392 "..ice age condition.." Replace with "..ice age conditions.."*

Thanks for the correction. Done.

*Line 402 "…cycles operates…" Replace with "…cycles operate…"*

Thanks for the correction. Done.

*Line 425 "U.S. American…" Replace with "American…"*

Done.

*Line 430, Line 433, Line 435, I think in each case "gram-calories per cm2." Should be "gram-calories per cm2 per minute."*

I follow Editor's suggestion. Done.

*Line 519 "…the expected help.." Replace with "..the required help…"*

Many thanks for the suggestion. Done.

*Line 530 "..was genuine…" Replace with "..was productive.."*

Many thanks for the suggestion. Done.

*Line 531 Delete "about"*

It's deleted.

*Line 595 Delete "after"*

It's deleted.

*Line 598 "…until 1970's." Replace with "..until the 1970's."*

Thanks for the correction. Done.

**Reply to General Comments of CC1**

I am grateful to Ön who kindly accepted to read and provide feedback on this current version of the paper. I hereby gratefully acknowledge his encouragement and friendship. Ön gave up some of his precious time to comment on this initial manuscript. I hope my replies below answer some of his comments/questions[1]:

1. The article gives the impression as the ice age concept is proposed by Agassiz. But, unlike Agassiz, the text should also give credit to Venetz, de Charpentier (see, Berger, 2012) or even to Hutton (see, Davies, 1968) and Playfair (see, Seylaz, 1962). Furthermore, mentioning Jens Esmark (Andersen, 1992, pp. 102) as the first scientist to propose an astronomical solution (even it is very primitive) would be better.

 I think that the point you raised is a sound one. It's a good idea to briefly highlight the contributions of the scientists you mentioned, just before proceeding to Agassiz. Frankly, a similar suggestion came from Jan Mangerud, who reached me via e-mail. Along with the articles you mentioned, I will also use the article he recommended[2] in the final version of my manuscript.

2. In the 6th section the astronomical parameters (lines between 201–254)are given with too much detail. They can be simplified. Furthermore, figures 3, 4 and 5 can be presented in a single figure.

I agree with your suggestion about combining the figures. I will also try to eliminate any unnecessary detail -especially in lines between 201–254.

3. The formulas between 2 to 9 distorts the integrity of the text. I believe these formulas and the paragraphs describing them should be rewritten from scratch just by referencing the original sources.

I agree with your comment. Additionally, this comment is largely in agreement with the first (Anonymous) and second referee's (Andreas Schmittner's) concerns. Consequently, the equations from (2) to (9) will not appear in its original form. Instead you will find equations explained descriptively. To enhance the accessibility of the text, I will state Milankovitch's equations in words.

4. In text it is sometimes written the earth and sometimes Earth . It should be consistent. If I were Ates, I would follow the recommendation of Sengoer (2017) and use it as the earth not Earth

 I will consider this suggestion as well.

**Line based recommendations of CC1 and My opinions/replies**

•Line 38 (L38): Change "our climate is subject to certain periodical changes" to "the earth's climate is subject to certain quasi-periodic changes"

Done.

• L41-42: Change "Whether periodically or not, the Earth has witnessed, and probably will continue to witness numerous glacial and interglacial periods." to "Whether periodic or not, the earth has witnessed, and probably will continue to witness numerous glacials and interglacials."

Done.

• L45: "than their previous ones?" or "than some other ones?"

I will follow your suggestion. Using  "than some other ones" is more appropriate.

• L49: I would delete the first sentence of this paragraph, starting with "There are many causes of major glaciation,...".

I agree with your suggestion.
* * *
[1] Ön's comments are colored red, Authors' replies are black.
[2] https://onlinelibrary.wiley.com/doi/10.1111/bor.12260

• L50: Consequently, I would change the sentence " Among all these, however, the main cause responsible to initiate an ice age period is plate tectonics." to "Without the feedback mechanisms initiated by plate tectonics, long term oscillations of climate (as illustrated in Figure 1) would be completely different."

Done. Thanks for the suggestion.

• L51 A comma after meteorologist.

OK

• L51 "this hypothesis", which hypothesis?

The whole paragraph has been revised in light of this comment. Thanks.

• L52 Change had to was.

Done

• L53: This sentence is odd. The first part, till the comma, the reader understands it as Wegener's theory revealed the mechanisms of ice ages. Maybe it can be, "He spent his time primarily in Greenland and his field research was mainly focused on continental drifts that led him to develop the revolutionary theory of plate tectonics, which brought a useful explanation for long term climate changes." Thanks for all the suggestions. The last five comments above are all related to one passage (L49-54). So accordingly the first paragraph of Sec.2 has been deleted and replaced as follows:

Without the feedback mechanisms initiated by plate tectonics, long term oscillations of climate (as illustrated in Figure 1) would be completely different. So to speak, the main cause responsible to initiate an ice age period is plate tectonics. Alfred Wegener (1880–1930), the German geologist and meteorologist, has laid the groundwork for this idea. He spent his time primarily in Greenland and his field research was mainly focused on continental drifts that later led Harry Hess (1906–1969) to develop the revolutionary theory of plate tectonics which brought a useful explanation for long term climate changes. The theory of plate tectonics was not peculiarly aimed to find out the mechanism of glacial periods, but it provided a useful framework for how these periods occur.

One more thing. I realized that I forgot to mention Hess in this initial manuscript. Thus, I hereby took the opportunity to mention his name in the passage.

• L58: Don't use "beneath", maybe "south" is better.

OK.

• L 81 Delete "has".

Done.

• L 82 Change "an extensive ice age about 350–250 million years ago, also the era when Pangaea existed. This can also be seen in (Fig .1), where one of the lowest points of the curve denotes this period." to "an extensive ice age about 350–250 million years ago, also the era when Pangaea existed (Fig .1)"

Thanks, I follow your suggestion.

• L93: Referencing a first year textbook, Lutgens et al. (2012)... I am not sure if it is a good idea. Furthermore, this is the second time up to now and these are direct quotations.

Yes, this is a fair comment. I will paraphrase the information in my own words.

• L124–L129: The story about Agassiz is not that innocent (Berger, 2012,pp. 109, 2nd column)

I acknowledge the comment of the reviewer. I will add extra information about the issue.

• L149: How would it be to change the wording here from "discovery" to "hypothesis"?

Thanks for the suggestion, but I prefer to keep this word as it is.

• L167: Comma after Milankovitch.

Done.

• L176: "Until the era of Milankovitch, the mainstream methodology on particular issues in geology was descriptive.", is this sentence necessary? Or, even true?

Thanks for the comment. I will rewrite or eliminate this sentence.

• L189–L193: "Milankovitch approached the problem was quite original.", but in this paragraph what you explain is similar to Croll's approach. What is original?

To clarify, what I have intended to say is that Milankovitch's approach is similar to Croll in a general sense. What differs Milankovitch from Croll is that he takes all relevant orbital elements (not only eccentricity and precession but also obliquity) into account Thank you for your comment, I will clarify the issue by emphasizing this point in the final manuscript.

• L217: There is no need to show the computational line. The second line can be given in text, of course without that much precision, such as e = 0 . 016.

OK.

• Fig3: Please note that it is today's condition.

OK

• L236: Are following sentences really necessary? "This tilt can be measured easily at solstices and equinoxes. In order to do that, it is sufficient to take the inverse tangent of the value which is found by dividing an object's height by its shadow length, at that particular time."

Thanks for the comment. Of course it is not necessary, but this is a short explanation or an extra piece of information which I see as harmless. So I prefer to keep this sentence as it is.

• L243: Yes, it is true that the precession of the equinoxes cycle lasts approximately 26,000 years. However, the climate is not affected by the sole effect of precession, but by the effect of the precession of the equinoxes modulated by the changes in the eccentricity of the earth's orbit. On average, its period is 21,700 yrs. Please see Berger (Table 1 in 1977).

Thanks for this comment. Here I am not considering the eff ect of precession on the climate. This passage merely intends to defi ne the precession of the equinoxes cycle.

• L245: Will Polaris once again be the North Star after 13,000 years or after 26,000 years? I suspect, there is a small mistake here.

Today the polar axis of the earth points to Polaris; after 13.000 yrs it will point to Vega, and after 13.000 yrs it will once again point to Polaris. Perhaps a clarification is required here. Thank you.

• L255-261: It would be appropriate if you give original references of these studies.

OK.

• L275: I am not sure, but would it be OK to add that Croll was aware of the continental distribution of the earth?

OK. Thank you for pointing this out to me.

• L321: "It is because all three cycles operates independently of each other.", I didn't understand the causality of being independent and being in a superposition of these cycles for a significant global climate change.

True. What I have in my mind is as follows: These cycles are neither singly nor jointly the direct cause of glaciation; instead they merely determine the crucial levels of insolation. Thus, glaciation (as well as inter-glaciation) process causally connected with the amount of solar insolation, not with the orbital cycles of the Earth. However, these cycles, incorporate as modulating elements of main causal factor (i.e. incoming solar radiation) that eventually influence global climate change.

• L323: "When the quantity of the heat decreases, a glacial period begins. In the opposite situation, when the quantity of heat increases, the global temperature significantly rises up and an interglacial period begins con sequently.", I don't think this is true. Insolation is not the marker of beginning of glacials and interglacials (cf. PAGES, 2016). I believe, these are all of Milankovitch's ideas or part of Milankovitch's model. They are not true for today. Therefore it should be explicitly emphasized.

True, I need to (and I will) stress this point in the final manuscript.

• L395: "This could be evaluated as an indication of a relatively cold summer for the northern hemisphere ."

I agree.

• L399: Is the following sentence correct? "Both scientists were already presented their results in a graph."

This sentence is deleted. The revised text is as follows: Soon after completing his work, Milankovitch sent this radiation curve graph to Koeppen. The graph must have been intriguing for Koeppen, as it was in agreement with the findings of two German geographers Albrecht Penck (1858–1945) and Eduard Brueckner (1862–1927). Both scientists were already presented their results in a graph (Fig. 8).Penck and Brueckner were researching on Alpine glaciers and about 15 years before Milankovitch's work, they identified four great glacial periods in Earth's history by examining successive gravels, plant remains and moraines in the European Alps (Anderson et al., 2013 p. 8). They displayed these different periods in a graph and named them chronologically as Guenz, Mindel, Riss, and Wuerm in their 1909 book 'Die Alpen im Eiszeitalter' (Fig. 8).

[Figure]

Kindly notice that the above figure (fig.8 with certain revisions its number most probably will change) is redrawn as a vector image. So, the image's initial poor quality will not remain.

• L418: Is it true with human effects? Please check this. See also Fig. 6 of Paillard (2010) for possible future scenarios.

Thanks for the comment. As you point out, the issue about human effects seems controversial. I will make additional statements to clarify this specific issue.

**Additional Changes:**

*(Abbreviation: in revised manuscript line [RM])*

- The word "mankind" is substituted with "humankind"(in revised manuscript line 34)
- The word "chapter" is substituted with "article" (RM 41).
- The conjunctions "are" & "or" deleted, "exists" & "and" inserted, respectively (RM 44).
- The first paragraph of second section is rewritten.
- Figure 2, remained same but in more high resolution version.
- The number "250" is substituted with "260" for consistency (fig.1).
- Additional reference added (RM 93).
- Additional reference added (RM 107).
- A new paragraph added (RM 121-136).
- Properly quoted from the original source material (RM 143-149).
- A new sentence added (RM 154-155).
- New paragraphs added to explain the arguments of Adhémar (RM 158-192).
- New paragraphs added to explain the arguments of Croll (RM 227-288).
- The word "in detail" is substituted with "and detailed" (RM 260-261).
- The word "papers" is substituted with "book *Kanon der Erdbestrahlung und seine Anwendung auf das Eiszeitenproblem*" , (RM 266-267)
- The word "*all*" is inserted (RM 286).
- Two new paragraphs (RM 295-306) and two new images (fig.3 & 4) are added.
- The word "approximately" is inserted (RM 326).
- New sentence added (RM 344-346).
- New paragraphs added (RM 361-385).
- "Adhémar actually thought it was 22.000 years" inserted in brackets (RM 388-389).
- "again, for Adhémar it was 11.000 years" inserted in brackets (RM 392).
- The word "Earth" inserted (RM 393).
- The word "idea" inserted (RM 394).
- The paragraph of second section is rewritten (RM 401-406).
- New paragraphs added (RM 416-464).
- Symbols deleted (RM 465-468).
- A new paragraph added (RM 505-517).
- The word "letter" is substituted for "mail" (RM 523).
- A new paragraph added (RM 535-547).
- Figure 5, remained same but in more high resolution version.
- "PFGQHKSF'" is substituted for "PFGHKSF'" (RM 555).
- A new paragraph added (RM 559-564).
- Figures 6 & 7 are remained same but displayed in more high resolution version.
- New references are added. In the alphabetical order: *Adhémar(1842), Agassiz(1838;1886; 1871/1949), Andersen (1992), Berger (2021), Bol'shakov et al. (2012), Croll (1864; 1875; 1890), Davies (1968), Evans (1887), Forbes (1853), Hay (1996), Hestmark (2018), Janc et al. (2020), Langley (1903), Murphy (1869), Paillard (2010), Seylaz (1962) and Thompson (2021).*